# Quantification of extremal dependence in spatial natural hazard footprints: independence of windstorm gust speeds and its impact on aggregate losses

Laura C. Dawkins[1]* and David B. Stephenson[1]

[1] College of Engineering, Mathematics and Physical Sciences, University of Exeter,

Exeter, UK

* E-mail: L.C.Dawkins@exeter.ac.uk

## Abstract

Natural hazards, such as European windstorms, have wide spread effects, causing insured losses at multiple locations throughout a continent. Multivariate extreme-value statistical models for such environmental phenomena must therefore accommodate very high dimensional spatial data, as well as correctly represent dependence in the extremes to ensure accurate estimation of these losses. Ideally one would employ a flexible model, able to characterise all forms of extremal dependence. However, such models are restricted to a few dozen dimensions, hence an a priori diagnostic approach must be used to identify the dominant form of extremal dependence. Current approaches for doing so are, however, also based on relatively low dimensional data.

Here, we present an approach for systematically exploring the dominant extremal dependence class in a very high dimensional spatial hazard field. In addition, we contribute a further, natural hazards relevant diagnostic by exploring the impact of extremal dependence misspecification on conceptual aggregate hazard loss estimation. These approaches are illustrated by application to a dataset of high dimensional historical European windstorm footprints (spatial maps of 3-day maximum gust speeds at $\sim$15000 locations).

We find there is little evidence of asymptotic extremal dependency in windstorm footprints. Furthermore, empirical extremal properties and conceptual losses are shown to be well reproduced using Gaussian copulas but not by extremally-dependent models such as Gumbel copulas. It is conjectured that the lack of asymptotic dependence is a generic property of turbulent flows, which may extend to other spatially continuous hazards. These results motivate the potential of using Gaussian process (geostatistical) models for efficient simulation of hazard fields.

**Key Words**: Natural hazards; Windstorm footprint; Extremal dependence; Reinsurance; Copulas

# 1   Introduction

Multivariate statistical models are increasingly used to explore the spatial characteristics of natural hazard footprints and quantify potential aggregate losses. For example, such

models for European windstorms are used by academics and re/insurers to create catalogues of possible events, explore loss potentials, and benchmark synthetic events from atmospheric models (Bonazzi et al. 2012; Youngman and Stephenson 2016).

Natural hazards, such as European windstorms, have wide spread effects, often causing insured losses at many locations throughout a continent. Therefore, statistical models for such hazards must accommodate very high dimensional data in order to represent the full hazard domain. Moreover, since natural hazards are rare events in the tail of the distribution, these statistical models must also correctly represent the dependence in the extremes to ensure valid inference and, hence a realistic representation of the hazard's aggregate losses.

When modelling multivariate extremes, variables can be described as being either asymptotically dependent, where large values of the variables tend to occur simultaneously, or asymptotically independent, where the largest values rarely occur together (Coles et al., 1999). As noted by Wadsworth et al. (2017), examples of modelling joint extremes often assume asymptotic dependence in order to accommodate asymptotically justified extreme value max-stable models, potentially leading to over-estimation of the joint occurrences of extremes, if incorrect. This assumption is common in the field of natural hazard research. Coles and Walshaw (1994) used a max-stable model for the dependence in maximum wind speeds in different directions; Blanchet et al. (2009) to model snow fall in the Swiss Alps; Huser and Davison (2013) to model extreme rainfall and Bonazzi et al. (2012) to model windstorm hazard fields at pairs of locations in Europe. Indeed, Bonazzi et al. (2012) simply base this modelling assumption on being "in line with many examples found in the literature". Therefore, it is important to ask: how valid is this assumption of asymptotic dependence? And how much of an effect might a misspecification of extremal dependence have on the resulting hazard loss representation in the model?

Two approaches for exploring, and correctly representing, extremal dependence are present in the literature. These involve using either a flexible model, able to represent both forms of extremal dependence, or a set of diagnostic measures to identify extremal dependence class prior to fitting a model with the diagnosed form of extremal dependence.

There is a growing literature in the area of flexible models for extremal dependence, originating from the bivariate tail model of Ledford and Tawn (1996), varying in their

merits and limitations. Wadsworth and Tawn (2012) developed a spatial model, involving inverted max-stable and max-stable models, able to incorporate both forms of extremal dependence. This model, however, requires the estimation of a large number of parameters and is only able to transition between dependence classes at a boundary point of the parameter space. Following this, Wadsworth et al. (2017) explored more flexible transitions between extremal dependence classes and developed a model able to represent a wider variety of dependence structures, although limited to the bivariate case. Huser et al. (2017) went on to develop a flexible extension of the Wadsworth et al. (2017) model using Gaussian scale mixtures, in which the two asymptotic dependence regimes are smoothly bridged between, and estimated from the data. As noted by Huser and Wadsworth (2018), however, this model either makes the transition between dependence class at a boundary point of the parameter space (as in Wadsworth and Tawn 2012), or is inflexible in its representation of the asymptotic independence structure. Huser and Wadsworth (2018) presents the most recent advancement, in a flexible model able to capture both extremal dependence classes in a parsimonious manner, provide a smooth transition between the two cases and cover a wide range of possible dependence structures, all based on a small number of parameters.

While these models provide a great advantage in terms of flexibility and are growing in their applicability to higher dimensions, none are computationally feasible for very high-dimensional datasets (Huser and Wadsworth, 2018), as required for natural hazards modelling over a large spatial domain. Indeed, max-stable models for asymptotic dependence are limited in application to a few dozen variables due to the computational demand of existing fitting methods (de Fondeville and Davison, 2018). Hence, as noted by Huser and Wadsworth (2018), with the exception of the specific high-dimensional peaks-over-threshold model of de Fondeville and Davison (2018), truly high-dimensional inference for spatial extreme-value models has yet to be achieved.

As a result, when aiming to model very high-dimensional data, the alternative, a priori identification of extremal dependence class approach must be taken, and an appropriate model then selected based on this identification. For example the model of de Fondeville and Davison (2018) for asymptotic dependence or a geostatistical, multivariate Gaussian model for asymptotic independence.

A number of papers have developed and/or employed diagnostic measures to identify

the form of extremal dependence between variables, prior to model fitting. Ledford and Tawn (1996) and Ledford and Tawn (1997) developed a bivariate tail model in which one of the parameters, named the coefficient of tail dependence, is used within a diagnostic approach to help identify the bivariate extremal dependence class. Coles et al. (1999) introduced two extremal dependence coefficients, $\chi(p)$ and $\bar{\chi}(p)$, characterising the conditional probability of a pair of locations exceeding the same high quantile threshold $1 - p$, for which the asymptotic limit (as $p \to 0$) provides a diagnostic of bivariate extremal dependence. Bortot et al. (2000) used pairwise scatter plots and empirical estimates of $\chi(p)$ and $\bar{\chi}(p)$ to diagnose the form of extremal dependence present in a 3-dimensional dataset of sea surge, wave height and wave period in south-west England. They found evidence for asymptotic independence, and hence developed a multivariate Gaussian tail model for their data, derived from the joint tail of a multivariate Gaussian distribution with margins based on univariate extreme value distributions. Similarly, Eastoe et al. (2013) applied the coefficient of tail dependence, the $\chi$ and $\bar{\chi}$ measures, and the conditional extremes model of Heffernan and Tawn (2004) to estimate the form of extremal dependence in 3 hourly sea surface elevation maxima at 15 locations, identifying, in general, asymptotic dependence. Similarly, more recently, Kereszturi et al. (2015) employed the coefficient of tail dependence and $\chi$ and $\bar{\chi}$ measures within a comprehensive theoretical framework to assess extremal dependence of North Sea storm severity along four strips of 14 locations within the North Sea.

In all of the above examples these diagnostic approaches are applied to a relatively small number of locations. Here we present an approach for systematically exploring the dominant form of extremal dependence within a high dimensional natural hazards dataset. Specifically, we demonstrate this approach using a large ($\sim$6,000 events) and very high-dimensional dataset ($\sim$15,000 locations) of climate model generated European windstorm footprints.

We introduce the bivariate diagnostic measures of Ledford and Tawn (1996) and Coles (2001) in the context of our approach by initially using them to explore the bivariate extremal dependence in two pairs of locations within the European domain (London-Amsterdam and London-Madrid), and subsequently present an approach for systematically applying the same diagnostics throughout the high dimensional domain. We use the simple extremal dependence measures of Ledford and Tawn (1996) and Coles (2001)

as they are quick to compute and can therefore be calculated for many thousands of pairs of locations, important when exploring high dimensional data.

In addition, we contribute a further diagnostic, relevant for natural hazards modelling, by presenting an approach for exploring the impact of extremal dependence misspecification on conceptual aggregate hazard loss estimation. We use the Gaussian and Gumbel copula models, representing asymptotic independence and dependence respectively, to model pairs of locations, and quantify the discrepancy between modelled and observed joint conceptual losses. This approach is introduced for one central location, paired with all other locations in the high dimensional domain, and then extended to systematically explore the full domain. In the case where a combination of asymptotic independence and dependence is identified within the domain, this diagnostic is beneficial in understanding how using a model for one form of extremal dependence, necessary due to the high dimensionality of the data, effects this important natural hazards model output, hence providing further justification of the selected dependence model. The approaches presented in this paper could be used to explore extremal dependence and develop an appropriate multivariate statistical model for any alternative high-dimensional natural hazard dataset.

The remaining paper is organised as follows. The windstorm hazard dataset used throughout this paper, is described in Section 2. In Section 3 we introduce and apply the extremal dependence diagnostics of Ledford and Tawn (1996) and Coles et al. (1999), firstly to two pairs of locations and secondly to systematically explore the high-dimensional data domain. Section 4 describes our additional, natural hazards relevant, conceptual aggregate loss extremal dependence diagnostic approach. Section 5 contributes a physical explanation for the form of extremal dependence identified in the windstorm hazard fields, and finally, Section 6 concludes.

# 2 Data

The windstorm footprint data set used in this paper is the same as that used in Dawkins et al. (2016) and an extended version of the data set used in Roberts et al. (2014), consisting of the 6103 high resolution model generated windstorm footprints, for windstorm events that occurred within the European domain during the 35 extended winters (Oc-

tober - March) 1979/80 - 2013/14 (kindly provided by J. Standen and J. F. Lockwood at the Met Office).

The windstorm footprint is defined as the maximum three second wind gust speed (in ms$^{-1}$) at grid points in the region $15\,°\mathrm{W}$ to $25\,°\mathrm{E}$ in longitude and $35\,°\mathrm{N}$ to $70\,°\mathrm{N}$ in latitude over a 72 hour period centred on the time at which the maximum 925hPa wind speed occurred over land. The 925hPa wind speed is taken from ERA-interim reanalysis (Dee et al., 2011). The three second wind gust speed has a robust relationship with storm damage (Klawa and Ulbrich, 2003), and is commonly used in catastrophe models for risk quantification (Roberts et al., 2014). A 72 hour windstorm duration is commonly used in the insurance industry (Haylock, 2011), and is thought to capture the most damaging phase of the windstorm (Roberts et al., 2014).

These 6103 historical windstorm events have been identified using the objective tracking approach of Hodges (1995) and the associated footprints are created by dynamically downscaling ERA-Interim reanalysis to a horizontal resolution of 25km using the Met Office unified model (MetUM). As described by Roberts et al. (2014), the wind gust speeds are calculated from wind speeds in the MetUM model, based on a simple gust parameterisation $U_{gust} = U_{10m} + C\sigma$, where $\mathrm{U}_{10m}$ is the wind speed at 10 metre altitude and $\sigma$ is the standard deviation of the horizontal wind, estimated from the friction velocity using the similarity relation of Panofsky et al. (1977). The roughness constant $C$ is determined from the universal turbulence spectra and is larger over rough terrain.

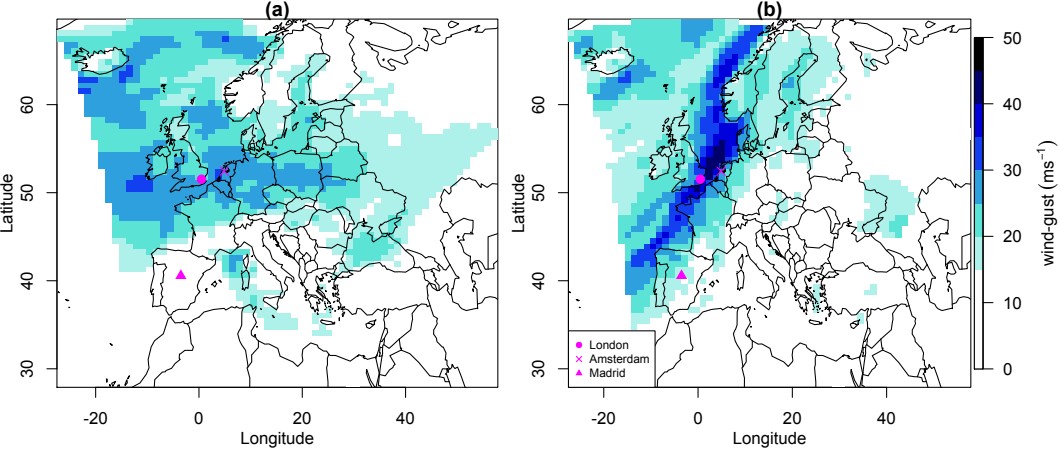

Figure 1: Hazard footprints for windstorms (a) Kyrill and (b) the Great Storm of October '87, with the location of the cities of London, Amsterdam and Madrid indicated.

The MetUM generated footprints for Kyrill ($17^{th} - 19^{th}$ January 2007) and the Great Storm of October '87 ($15^{th} - 17^{th}$ October 1987) are shown in Fig. 1. The variability in the intensity and location of extreme, damaging winds in these footprints demonstrate the potential importance of correctly modelling the spatial dependence between locations for realistically representing joint losses.

Using model generated windstorm footprints for representing historical storms has benefit in terms of spatial and temporal coverage, however these estimated maximum wind-gust speeds will inevitably differ from the those observed at nearby weather stations. For example, as noted by Roberts et al. (2014), several alternative methods for parameterising wind gust speeds are available (see Sheridan (2011) for a review), which can lead to large differences in estimated gusts (10-20ms$^{-1}$). The validity of simplistic gust parameterisation stated above was evaluated by Roberts et al. (2014), who found an overestimation in the effect of surface roughness at stations greater than $\sim 500$ metre altitude, leading to an underestimation in MetUM modelled extreme winds in these locations. In addition, Roberts et al. (2014) identified a slight underestimation in extreme wind gust speeds greater than $\sim 25$ms$^{-1}$, found to be due to a number of mechanisms including the underestimation of convective effects and strong pressure gradients, leading to the underdevelopment of fast moving storms (Roberts et al., 2014).

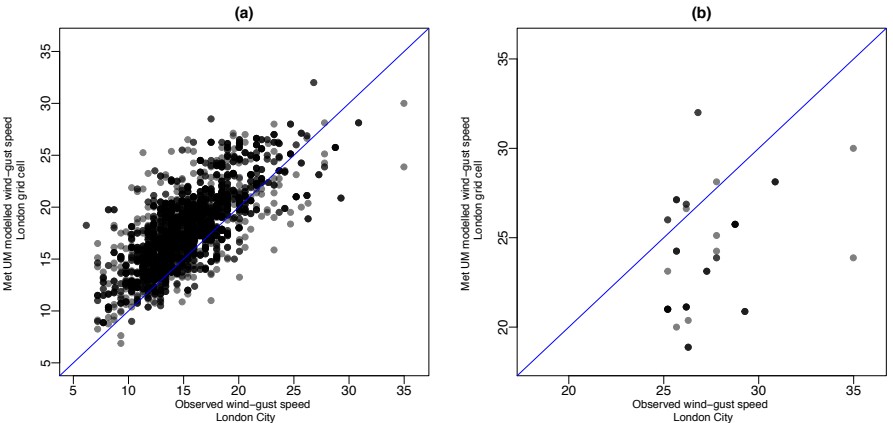

Figure 2: (a) The relationship between MetUM windstorm footprint wind gust speeds in the London grid cell and the corresponding observed wind-gust speeds at the London City weather station within the Global Summary Of the Day dataset, and (b) the same relationship for the 50 must extreme windstorm events at the London City weather station. In both plots the line $y = x$ has been added for reference (blue).

To explore the possible discrepancy in the MetUM wind gust speed data relevant for this study, we extract daily maximum observed wind gust speed recorded at the London City weather station (the station located within the London grid cell used throughout this study) from the Global Summary Of the Day (GSOD) data repository, and, for each of the 6103 windstorm events in our dataset, find the maximum observed gust in the 3 day period centred on the same date as in the MetUM model generated footprints. A comparison of the observed and MetUM modelled footprint wind gusts in London is presented in Fig. 2 (a), indicating a general overestimation in modelled wind-gust speeds below $25\text{m}^{-1}$ and a slight underestimation for wind-gust speeds above $25\text{m}^{-1}$, reflecting the findings of Roberts et al. (2014). Figure 2 (b) presents this same relationship for the 50 most extreme events in the observed dataset, highlighting this underestimation of modelled extreme wind-gust speeds. Indeed, the root mean squared difference between the observed and modelled footprint wind-gust speeds for these 50 extreme events is $4.57\text{ms}^{-1}$, giving an indication of the model uncertainty in representing extreme windstorm footprint wind-gust speeds.

The discrepancy in model generated wind-gust speeds compared to the observations could lead to differences in results, namely the identification of the extremal dependence class between locations. To explore this possibility we repeat the empirical analysis in Section 3 (Fig. 4) for GSOD data at London City and Amsterdam Schiphol Airport, shown in Figure 1 in the Supplementary Material. We find that for this pair of locations, the weather station and MetUM data have very similar relationships in the extremes, with the weather station data being slightly less dependent, therefore not changing the conclusions of the analysis.

# 3 Extremal dependency

As a motivating example, the bivariate dependence in windstorm footprint wind gust speeds for London paired with Amsterdam and Madrid are presented in Figures 3 (a) and (c) respectively. These three locations are shown in Fig. 1, and these two pairings are chosen because of their contrasting separation distances, and hence degrees of dependence (as shown in Fig. 2 in the Supplementary Material). These scatter plots show a greater degree of dependence between London and Amsterdam compared to London and Madrid.

<sub>230</sub> Indeed, multiple windstorms have losses occurring in London and Amsterdam at the same

<sub>231</sub> time, when loss is associated with wind gust speeds exceeding the 99% quantile at a given

<sub>232</sub> location, characterised by the top right-hand corner of each plot in Fig. 3. However, does

<sub>233</sub> this level of dependence between London and Amsterdam necessarily suggest asymptotic

dependence?

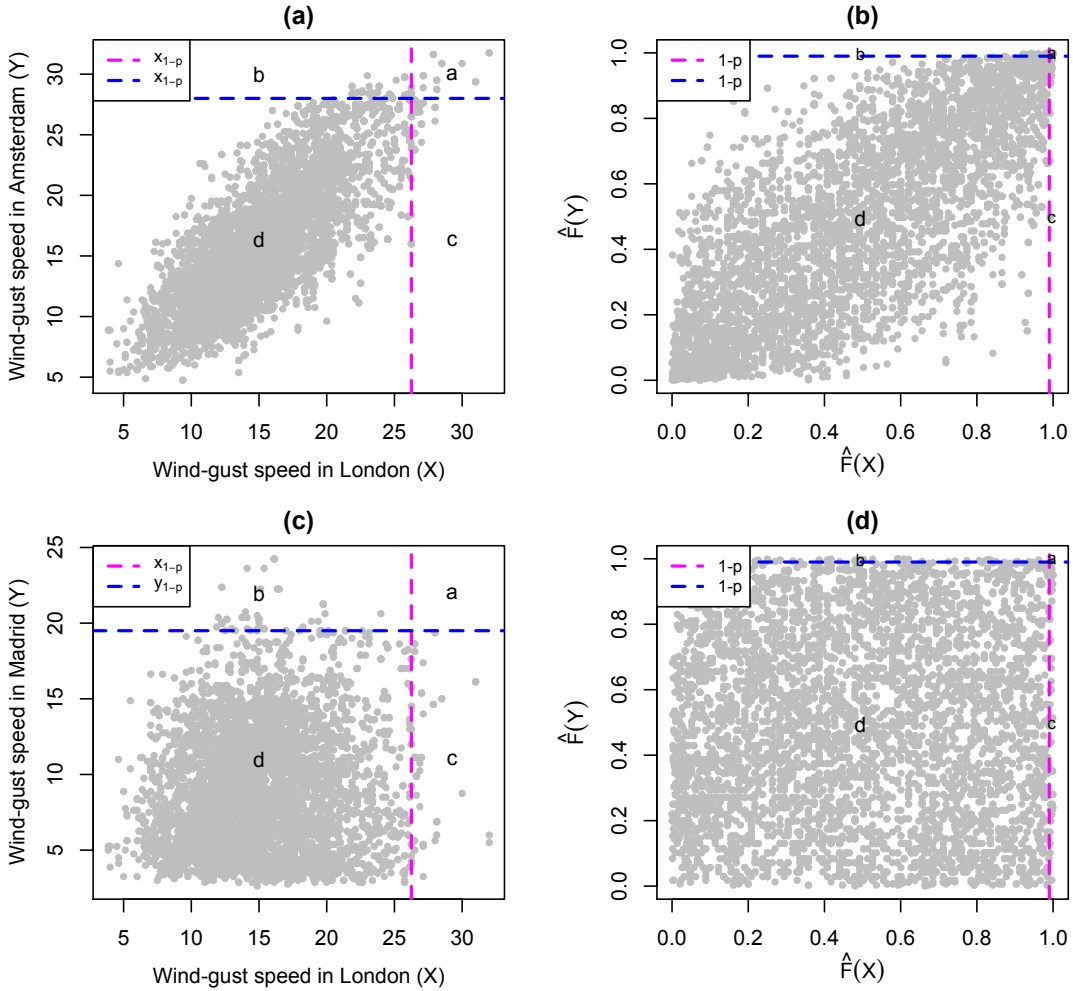

Figure 3: Scatter plot comparing historical windstorm footprint wind gust speeds $(\text{ms}^{-1})$ in London paired with (a) Amsterdam and (c) Madrid, and empirical copula plots for London paired with (b) Amsterdam and (d) Madrid. Dashed lines show the 99% quantile of wind gust speed at each location, and labels a-d represent the number of points in each section of each plot, related to being above or below these high quantile thresholds.

<sub>234</sub>

<sub>235</sub> Let the $n \times 2$ variable $(X, Y)$ represent the wind gust speeds associated with the

<sub>236</sub> $n = 6103$ windstorm events at any given pair of locations within the European domain,

e.g. London and Amsterdam. The bivariate relationship between $X$ and $Y$ can be represented by two components, the marginal distributions of each variable, and their joint dependence. The dependence component of the relationships shown in Fig. 3 (a) and (c) can therefore be isolated by, for each location, transforming wind gust speeds associated with each of the windstorm events, e.g. $X_i$ for $i = 1, ..., n$, to uniform margins using the estimator of the empirical distribution function ($\frac{1}{n} \sum_{j=1}^{n} \mathbb{1}_{X_j \leq X_i}$), shown in Fig. 3 (b) and (d) respectively. This is known as the empirical copula.

## 3.1 Diagnostic measures

The degree of conditional dependence between locations, at a specified high quantile threshold, $1 - p$, can be explored, based on the empirical copula, using the Extremal Dependence Coefficients, $\chi(p)$ and $\bar{\chi}(p)$, introduced by Coles et al. (1999), and the asymptotic limit of these measures, as $p \to 0$, classifies the class of bivariate extremal dependence as either *asymptotically dependent* or *asymptotically independent*. These measures are defined as,

$$\chi(p) = \Pr(Y > y_{1-p} | X > x_{1-p}) = \frac{\Pr(Y > y_{1-p}, X > x_{1-p})}{p}, \tag{1}$$

where $x_{1-p}$ and $y_{1-p}$ are the $(1-p)^{th}$ quantiles of $X$ and $Y$ respectively, $0 \leq \chi(p) < 1$ for all $0 \leq (1 - p) \leq 1$, and,

$$\bar{\chi}(p) = \frac{2\log(\Pr(X > x_{1-p}))}{\log(\Pr(X > x_{1-p}, Y > y_{1-p}))} - 1 = \frac{2\log(p)}{\log(\chi(p)p)} - 1 = \frac{\log(p) - \log(\chi(p))}{\log(p) + \log(\chi(p))}, \tag{2}$$

where $-1 \leq \bar{\chi}(p) < 1$ for all $0 \leq (1 - p) \leq 1$. Hence, if $\lim_{p\to 0} \chi(p) = \chi(0) > 0$, $\lim_{p\to 0} \bar{\chi}(p) = \bar{\chi}(0) = 1$, and the pair $(X, Y)$ are said to be asymptotically dependent with strength $\chi(0)$. If instead $\chi(0) = 0$, and hence, $\bar{\chi}(0) < 1$, the pair are said to be asymptotically independent, and the non-vanishing measure $\bar{\chi}(0)$ represents the strength of this non-asymptotic dependence.

As an initial exploration of bivariate extremal dependence class between variables, these conditional probability measures can be calculated empirically over a range of

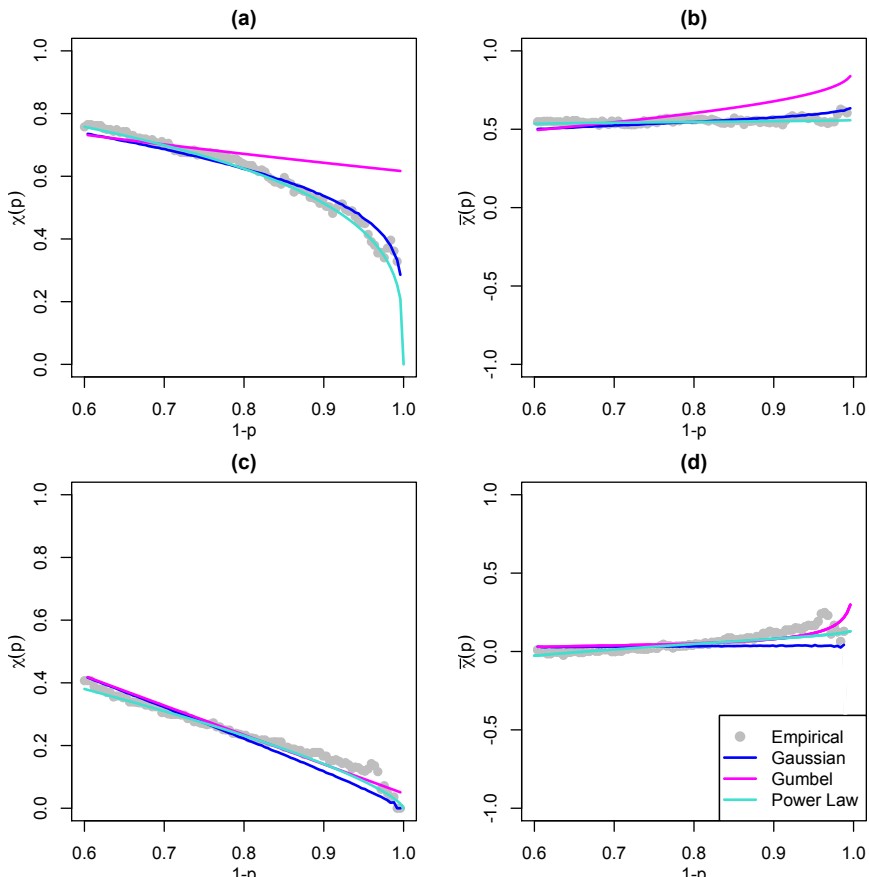

Figure 4: Extremal dependence measure $\chi(p)$, for $p \in [0, 0.4]$, for windstorm footprint wind gust speeds in London paired with (a) Amsterdam and (c) Madrid, and dependence measure $\bar{\chi}(p)$, for $p \in [0, 0.4]$, for windstorm footprint wind gust speeds in London paired with (b) Amsterdam and (d) Madrid, calculated empirically and based on the Gaussian, Gumbel and Power Law bivariate dependence functions, as defined in Table 1.

quantile thresholds, as shown in Fig. 4 for windstorm footprint wind gust speeds in London paired with Amsterdam and Madrid. These empirical estimates are calculated as functions of the counts (a,b,c,d) in Fig. 3, as defined in Table 1. Based on these empirical estimates, for both pairs of locations, $\chi(p) \to 0$ and $\bar{\chi}(p) < 1$ as $p \to 0$, suggesting asymptotic independence.

Here, however, and as in all datasets of environmental phenomena, the rarity of very extreme events makes it impossible to empirically quantify the asymptotic limits $\chi(0)$ and $\bar{\chi}(0)$, necessary for extremal dependence class identification. To overcome this, Ledford and Tawn (1996) developed a bivariate tail model, able to characterise both classes of extremal dependence, which when fit to a bivariate random variable can be used to predict the asymptotic limit of the conditional probability measures and hence give an estimate of the class of extremal dependence, based on the sub-asymptotic evidence in the data and the assumption that the model can be extrapolated to asymptotic levels.

As in Ledford and Tawn (1996), let $Z_X$ and $Z_Y$ denote $X$ and $Y$ transformed to unit Fréchet margins respectively, that is $\Pr(Z_X \leq z) = \Pr(Z_Y \leq z) = \exp(-1/z)$. Then the joint survivor function for $Z_X$ and $Z_Y$, above some large quantile threshold $z_{1-p}$, takes the form,

$$\Pr(Z_X > z_{1-p}, Z_Y > z_{1-p}) \sim \mathcal{L}(z_{1-p})p^{1/\eta}, \tag{3}$$

where $p = \Pr(Z_X > z_{1-p}) = \Pr(Z_Y > z_{1-p})$, $\frac{1}{2} \leq \eta \leq 1$ is a constant and $\mathcal{L}(z_{1-p})$ is a slowly varying function as $p \to 0$. Based on this power law model, as shown by Coles et al. (1999),

$$\chi(p) \sim \mathcal{L}(z_{1-p})p^{1/\eta-1},$$
$$\bar{\chi}(p) = \frac{2\log(p)}{\log(\mathcal{L}(z_{1-p})) + \frac{1}{\eta}\log(p)} - 1,$$
$$\to 2\eta - 1 \quad \text{as } p \to 0.$$

Hence, the parameter $\eta$, named the coefficient of tail dependence by Ledford and Tawn (1996), characterises the nature of the extremal dependence. When $\eta = 1$, $\chi(0) = \mathcal{L}(z_{1-p})$ and $\bar{\chi}(0) = 1$, hence the pair $(X, Y)$ are asymptotically dependent of degree $\mathcal{L}(z_{1-p})$.

Alternatively, if $\eta < 1$, $\chi(0) = 0$ and $\bar{\chi}(0) = 2\eta - 1$, and the pair are asymptotically independent with non-asymptotic dependence of degree $2\eta - 1$.

For a given pair, e.g. wind gust speeds in London and Amsterdam, the Ledford and Tawn (1996) model is fit to the joint survivor function along the diagonal, equivalent to the univariate distribution of $T = \min\{Z_X, Z_Y\}$, known as the structure variable, which has length $n$. Using the stable two parameter Poisson process representation of $T$, presented by Ferro (2007), who employed the Ledford and Tawn (1996) model for the verification of extreme weather forecasts, the exceedance of $T$ above a high threshold $w$ has the form,

$$\Pr(T > t) = \frac{1}{n}\exp\left[-\left(\frac{t - \alpha}{\eta}\right)\right] \quad \text{for all } t \geq w, \tag{4}$$

with location parameter $\alpha$ and scale parameter $0 < \eta \leq 1$, equivalent to $\eta$ in Eqn. (3), estimated by maximum likelihood (Ferro, 2007).

We fit this model to the pairs London-Amsterdam and London-Madrid, requiring the specification of the high threshold, $w$, above which the Poission process model is fit. As discussed by Ferro (2007), this threshold selection is a trade-off between being low enough that enough data is attained to ensure model precision, but high enough that the extreme-value theory that motivates the model provides accurate estimates, suggesting we should select the lowest level at which the extreme-value approximations are acceptable (Ferro, 2007). In a similar way to choosing the appropriate threshold when fitting a Generalised Pareto Distribution (see Coles 2001), empirical diagnostic plots can be used to inform this selection. For example parameter stability plots, in which the estimated model parameters and mean excess should be constant above the chosen high threshold; and quality of fit plots, in which for this model, the transformed excesses, $(Z - w)/\eta$, should be exponentially distributed if an appropriately high threshold has been chosen (see Ferro (2007) for more details).

Here, the 85% quantile of the structural variable $T$ is selected, based on these diagnostic plots (examples of these plots for London-Amsterdam are presented in Fig. 3 in the Supplementary Material). This choice is similar to the 0.88% and 0.9% thresholds selected in the applications of Ferro (2007) and Ledford and Tawn (1996) respectively. Based on this choice of $w$, $\eta = 0.78 < 1$ for London-Amsterdam and $\eta = 0.58 < 1$ for London-Madrid, indicating asymptotic independence for both pairs of locations. This

is further demonstrated in Figure 4 which shows how the Ledford and Tawn (1996) model, referred to as the Power Law model, calculated as in Table 1, represents the the conditional dependence measures $\chi(p)$ and $\bar{\chi}(p)$ as $p \to 0$, for London-Amsterdam and London-Madrid.

In addition, as a comparison (included in Fig. 4), alternative parametric bivariate dependence models known as the Gaussian and Gumbel copulas, can be used to model the pair $(X, Y)$ to give further indication of the extremal dependence class present.

The Gumbel bivariate copula model characterises asymptotic dependence with the degree of dependence quantified by parameter $r$. For each pair of locations, this parameter is estimated via maximum likelihood using the `copula` R package. The Gaussian bivariate model characterises asymptotic independence with dependence parameter $\rho$, here represented by the Spearman's rank correlation coefficient. Both models are fit to the full bivariate data pair, as presented in Fig. 3. For the Gumbel model the data is transformed to uniform margins using the empirical distribution function. The same transformation is made for the Gaussian model, followed by a transformation to Gaussian margins using the standard normal distribution function. The parametric forms of $\chi(p)$ and $\bar{\chi}(p)$ for these two opposing models are shown in Table 1. In Fig. 4, the Gumbel model is calculated as in Table 1, however, since the closed form definition for the Gaussian model in Table 1 only holds for the limit $p \to 0$, for this model $\chi(p)$ and $\bar{\chi}(p)$ are estimated as the median of 1000 parametric bootstrap simulations from the associated bivariate normal distribution.

For both pairs of locations in Fig. 4, all three parametric bivariate dependence models indicate asymptotic independence, since for the Power Law model $\chi(0) = 0$ and $\bar{\chi}(0) < 1$, the Gaussian model matches closely with the empirical estimates and the Power Law model, and the Gumbel model overestimates the conditional probability of joint extremes.

As a final diagnostic, analogous to that used by Ledford and Tawn (1996, 1997), the coefficient of tail dependence can be estimated for a range of high thresholds, $w$, to explore the sensitivity of the parameter estimate to this choice. As in Ledford and Tawn (1996, 1997), here this diagnostic observes the proportion of time $\eta = 1$ is within the profile likelihood confidence interval for $\eta$, when estimated using $w$ in the interval of $0.5 - 1$ quantile of $T$. The pair $(X, Y)$ are said to be asymptotically dependent if $\eta = 1$ is contained within these confidence intervals for a majority of the range of $w$,

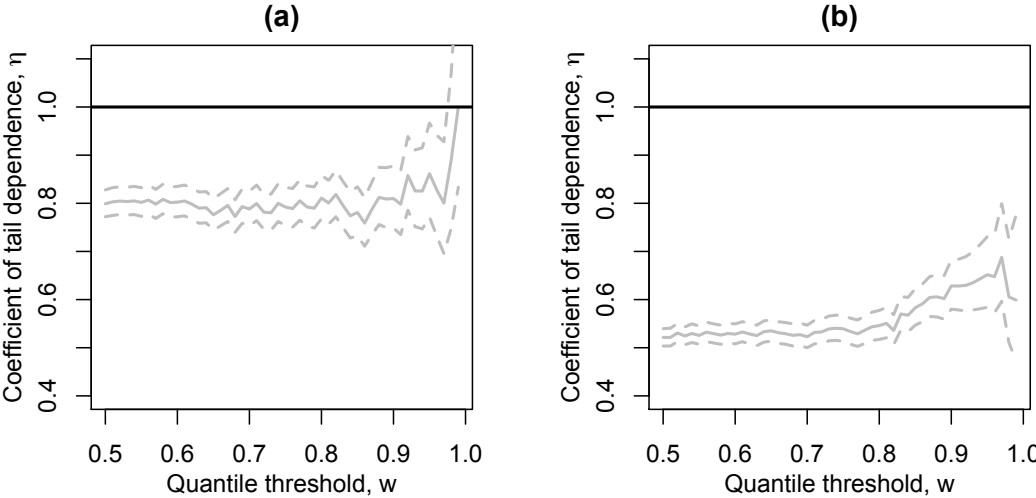

Figure 5: Diagnostic plots of maximum likelihood estimates (solid) and 95% profile likelihood confidence intervals (dashed) of $\eta$, in Eqn. (4), for threshold $w$ in the range of the $0.5 - 1$ quantile of $T$, for London paired with (a) Amsterdam and (b) Madrid.

and asymptotically independent otherwise. This exploration is presented for London paired with Amsterdam and Madrid in Fig. 5, providing further evidence of asymptotic independence for both pairs, based on this criterion.

## 3.2   Extending to high dimensions

We now present an approach for extending the quick-to-calculate coefficient of tail dependence diagnostic approach presented above to systematically explore the dominant extremal dependence class across locations in a high dimensional hazard field, demonstrated by application to our windstorm footprint data set.

We first take a stratified (based on the distribution of locations over longitude and latitude) sample of 100 locations within the European domain. One such sample is shown in Fig. 6 (a). Since the extremal dependence is likely to decrease with increasing separation distance (Wadsworth and Tawn, 2012) and we hope to understand if asymptotic independence is dominant and hence present at small separation distances, for each of these 100 locations, we estimate the coefficient of tail dependence, $\eta$ (and the associated 95% profile likelihood confidence interval) when paired with the 100 nearest locations within the full domain. Figure 6 (b) demonstrates how the 100 nearest locations are geographically

distributed for one such sampled location in our windstorm footprint dataset. For each pairing, the coefficient of tail dependence is calculated using w as the 0.9 quantile threshold of the structure variable, found to ensure stable estimates of $\eta$ using diagnostic plots as in Fig. 6 (c). The estimated $\eta$ parameters and confidence intervals for these $100\times100$ pairs of locations are plotted against separation distance to explore how, throughout the domain, $\eta$ varies at small separation distances and changes with increasing separation distance, shown in Fig. 6 (d). The parameter estimate related to the pair of locations in pink and blue in Fig. 6 (b), is shown in pink. This method is repeated many times with 10 such repetitions shown in Fig. 4 of the Supplementary Material, showing very similar results.

Figure 6 (d) shows that for small separation distances (<180 km) a proportion of pairs of locations have coefficients of tail dependence parameter, $\eta$, estimates close to 1, with $\eta = 1$ within the confidence interval, indicating asymptotic dependence. Within the range (0-50 km) 69% of pairs of locations exhibit this behaviour, however this proportion reduces rapidly as separation distance increases, to 30% for locations separated by (50-100 km), 13% for locations separated by (100-150 km) and 3% for locations separated by (150-200 km). Hence, while there is evidence of asymptotic dependence for some locations in close proximity, even at very small separation distances (50 km) a larger proportion of locations exhibit asymptotic independence. Indeed, here and in Fig. 4 of the Supplementary Material, beyond a separation distance of approximately 200km the coefficient of tail dependence parameter estimates drop well below 1, indicating asymptotic independence. Therefore, since separation distances within the domain extend to up to 3500km, we conclude that asymptotic independence is the dominant extremal dependence structure across the spatial domain.

It is important to consider the validity of representing even this small proportion of asymptotically dependent pairs of locations incorrectly as asymptotically independent. To explore this, Bortot et al. (2000) carried out a simulation study in which they fit the Gaussian, Ledford and Tawn (1996) and Gumbel models to bivariate data simulated from three parent populations with different classes of extremal dependence. They conclude that, for asymptotically independent parent populations the Gaussian copula is able to provide accurate inferences for tail probability estimates, out performing the Gumbel copula model, and even for asymptotically dependent parent populations, the estimation

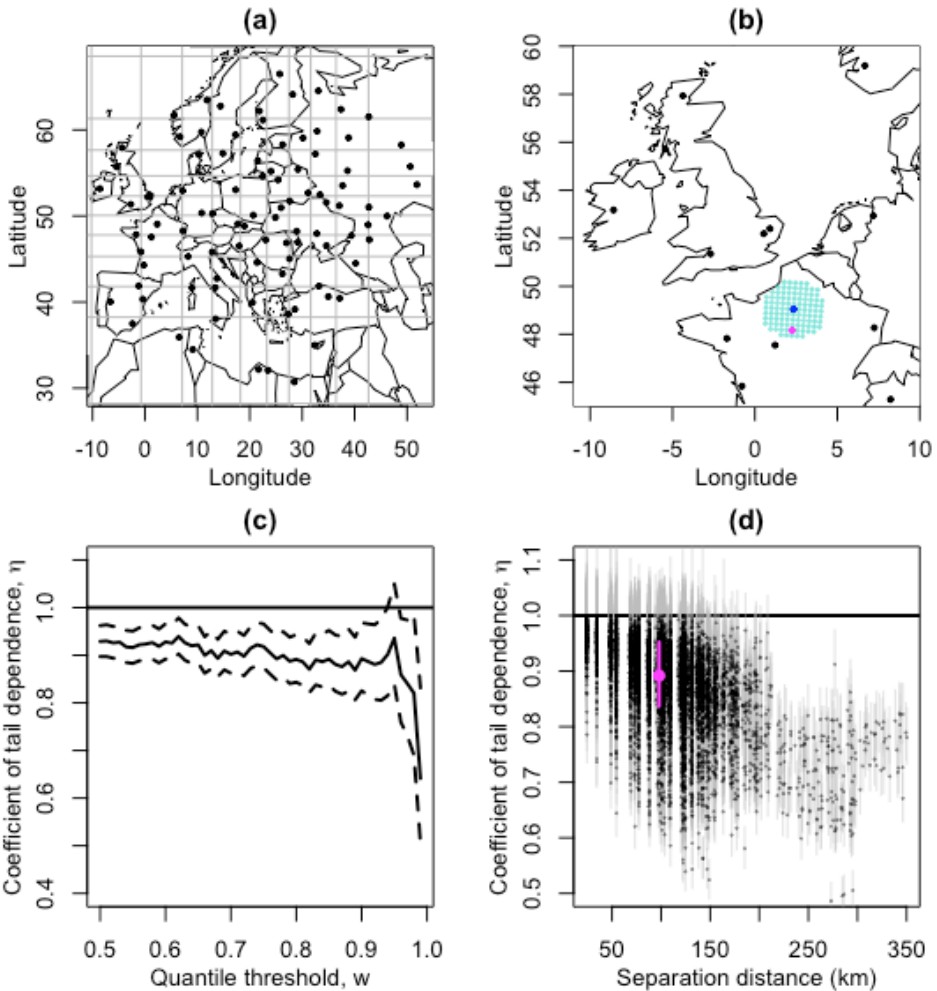

Figure 6: (a) A stratified (based on the distribution of locations over longitude and latitude) sample of locations within the European domain, with stratified grid shown in grey; (b) a demonstration of the 100 nearest locations [turquoise] to one of these sampled locations [blue], with one such point selected at random [pink]; (c) the coefficient of tail dependence diagnostic plot (as in Fig. 5) for wind gusts at the blue location paired with the pink location; (d) the coefficient of tail dependence (estimated using w as the 90% quantile threshold of the structure variable) and 95% profile likelihood confidence intervals, for each of the 100 sampled locations paired with their 100 nearest locations in the full domain, plotted against separation distance in kilometres, with the estimate based on the pair of locations in (b) and (c) added in pink.

error of the Gaussian copula model was deemed to be acceptably small. This suggests that, when data dimensionality prohibits the use of flexible extremal dependence models, such as Huser and Wadsworth (2018), and asymptotic independence is found to be the dominant extremal dependence structure across the spatial domain, using an asymptotically independent model, such as the Gaussian tail model, is preferable over using a model for asymptotic dependence throughout the domain. In Section 4 we present a further, natural hazards relevant, diagnostic approach for further validating this, based on estimates of the aggregate natural hazard losses.

# 4   A conceptual loss diagnostic approach

We now contribute an additional, natural hazards relevant diagnostic approach for exploring the dominant extremal dependence class, providing further justification of the selected dependence model. We define a conceptual hazard loss function and explore the impact of misspecifying the extremal dependence class on aggregate hazard loss estimation, using the Gaussian and Gumbel copula models previously introduced. We present this approach initially based on one central location (London), and then demonstrate how this can be extended to systematically explore a high dimensional hazard field.

Similar to other natural hazard loss models, in the absence of confidential insurance industry exposure and vulnerability information, it has become common in the literature to define conceptual windstorm loss as a function of the footprint wind gust speeds (see Dawkins et al. (2016) for a review). While these conceptual windstorm loss functions vary in the detail of their composition, it is possible to express most in a general form, for the pair $(X, Y)$, as:

$$L(X, Y) = g[V(X)e(X)H\{X - U(X)\} + V(Y)e(Y)H\{Y - U(Y)\}] \qquad (5)$$

where $V$ is a function the wind gust speeds characterising the magnitude of the hazard, $e$ represents exposure (e.g. population density), $U$ quantifies a high threshold of the wind gust speed above which losses occur, $H$ is a Heaviside function such that $H\{m\} = 1$ if $m > 0$ and $H\{m\} = 0$ otherwise, and $g$ is an additional function applied in some cases to reduce skewness. For example, in the widely used and rigorously validated conceptual loss function of Klawa and Ulbrich (2003), $V(X) = (X - x_{0.98})^3$, $U(X) = x_{0.98}$ (where

$x_{0.98}$ is the 98% quantile of $X$) and $e(X)$ is represented by the population density at the location (with equivalent expression for $Y$), while Cusack (2013) used a loss function in which $V(X) = (X - x_{0.99})^3$, $U(X) = x_{0.99}$, the 99% quantile of $X$, and $g[\cdot] = \sqrt[3]{\cdot}$. See Table 2.1 in Dawkins (2016) for a summary of previously published conceptual loss functions in terms of the components of Eqn. 5.

More recently, Roberts et al. (2014) presented an exploration of the success of a number of these conceptual windstorm loss functions in representing insured loss throughout the European domain, based on the same data set as in this study, with the aim of developing a method for selecting extreme storms for the eXtreme WindStorms (XWS) catalogue. While there is much published work on the existence of a relationship between loss severity and the magnitude of the wind, in particular the cubed excess wind as used in the loss functions of Klawa and Ulbrich (2003) and Cusack (2013), Roberts et al. (2014) found that a conceptual loss function representing just the area in which the windstorm footprint exceeds a high loss threshold (i.e. $V(X) = 1$ and $e(X) = 1$ in Eqn. 5) to be more successful at characterising a subset of extreme windstorms known to have caused large insured losses. It should be noted however, that this exploration did not include population density within the Klawa and Ulbrich (2003) loss function, and was therefore not a direct comparison of this measure. In addition, an alternative subjectively selected subset of extreme storms may have given an alternative result, and the success of this simplistic 'areal frequency of loss' function in representing losses in this climate model generated data set of windstorm footprints may be due to its relative insensitivity to errors in other components of the loss estimates, such as estimated gusts, and may not perform as well as other loss functions if applied to wind gust observations.

However, following the results of Roberts et al. (2014) in the context of this data set, and in line with Dawkins et al. (2016), within this study we propose a similar threshold exceedance conceptual loss function. Roberts et al. (2014) used an exceedance threshold of 25ms$^{-1}$ while Dawkins et al. (2016) used a threshold of 20ms$^{-1}$, as is commonly used by German insurance companies (Klawa and Ulbrich, 2003). Here, similar to Klawa and Ulbrich (2003) and Cusack (2013), we propose a locally varying wind gust speed quantile threshold, accounting for local adaptation to varying wind intensity. We find that the 99% quantile of windstorm footprint wind gust speed is in excess of the commonly used 20ms$^{-1}$ loss threshold for most land locations in Europe, with a higher loss threshold

used in regions where stronger winds occur (as shown in Figure 5 in the Supplementary Material).

Since, for a given storm event, $V(X)e(X)$ and $V(Y)e(Y)$ in Eqn. (5) are constants, this equation can be simplified to:

$$L(X,Y) \propto C_X H\{X - U(X)\} + C_Y H\{Y - U(Y)\} \tag{6}$$

where $C_X = V(X)e(X)$ and $C_Y = V(Y)e(Y)$. In our case $C_X = C_Y = 1$, and $U(X) = x_{0.99}$, $U(Y) = y_{0.99}$, the 99% quantiles of $X$ and $Y$ respectively. Therefore, while in this study we use just one conceptual loss function in which the magnitude of the loss is always equal to 1, it is simple to adapt the following analysis to accommodate alternative loss functions in which the size of the loss is included as a function of the excesses of the natural hazard, by incorporating a model for the marginal distribution of hazard at each location. This would be an interesting area of future exploration within this windstorm footprint application, beyond the scope of this analysis.

The probability mass function of the bivariate conceptual loss function can easily be obtained in terms of the Extremal Dependence Coefficient, $\chi(p)$, by considering the joint probability of $(X,Y)$ in each of the quadrants shown in Fig. 3:

$$\Pr(L(X,Y) = C_X + C_Y) = \chi(p)p,$$
$$\Pr(L(X,Y) = C_X) = \Pr(L(X,Y) = C_Y) = 2(1 - \chi(p))p,$$
$$\Pr(L(X,Y) = 0) = 1 + p(\chi(p) - 2),$$

This indicates that the success of a given model in representing the bivariate conceptual loss for the pair $(X,Y)$ closely relates to its characterisation of $\chi(p)$, where here $p = 0.01$, and hence the extremal dependence between $X$ and $Y$.

To compare how well the Gaussian and Gumbel models represent our empirical bivariate conceptual loss function we can therefore compare estimates for $\chi(p)$ and $\bar{\chi}(p)$ for our specified loss threshold $p = 0.01$, calculated based on each model, with those calculated empirically (as in Table 1). We present the resulting difference in these estimates for London paired with all other land locations in the European domain in Fig. 7.

Figure 7 demonstrates how, for London paired with all other locations, the Gaussian model is able to represent empirical $\chi(0.01)$ well throughout the domain. Conversely

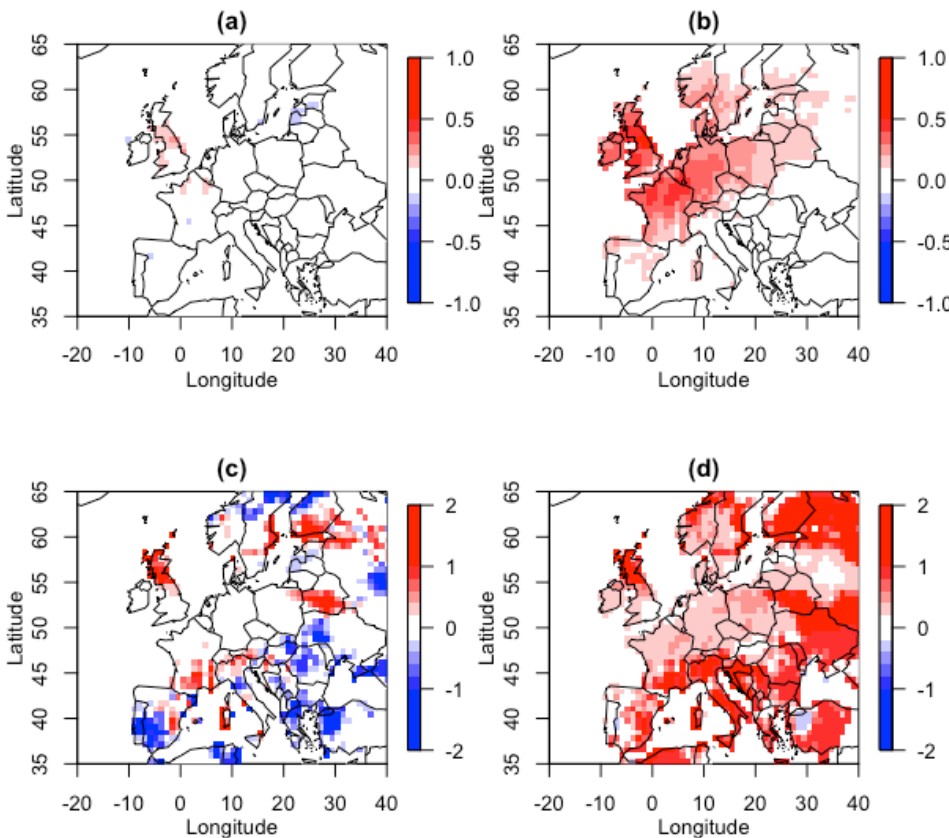

Figure 7: The difference between empirical and modelled $\chi(0.01)$ for (a) the Gaussian model and (b) the Gumbel model, and the difference between empirical and modelled $\bar{\chi}(0.01)$ for (c) the Gaussian model and (d) the Gumbel model, for London paired with all other locations over land.

the Gumbel model greatly over estimates $\chi(0.01)$ for all pairs of locations with non-zero empirical $\chi(0.01)$, bar the neighbouring grid cell. However, this neighbouring grid cell is also well represented by the Gaussian model. The Gaussian model reproduces $\bar{\chi}(0.01)$ well for locations within a small to medium separation distance from London, with this distance being greater in the West-East direction, reflecting the common path of storms over Europe (Hoskins and Hodges, 2002). The Gaussian model over and under estimates $\bar{\chi}(0.01)$ for far away locations. This discrepancy is most likely due to the very small sample of joint extremes at these pairs of locations making estimates of $\bar{\chi}(0.01)$ highly uncertain. The Gumbel model greatly overestimates $\bar{\chi}(0.01)$, for all locations, except again for those locations in very close proximity to London. This discrepancy in the Gumbel model is likely due to a misspecification of asymptotic dependence between most locations, resulting in an overestimation of the conditional dependencies in the extremes.

As well as being relevant for representing the probability mass function of the bivariate conceptual loss function, $\chi(p)$ can also be shown to characterise the conditional expectation of joint loss:

$$\mathbb{E}(L(X,Y)) = (C_X + C_Y)\chi(p)p + C_X(1 - \chi(p))p + C_Y(1 - \chi(p))p = (C_X + C_Y)p,$$

$$\Rightarrow \mathbb{E}(L(X,Y)|L(X) = C_X) = (C_X + C_Y)\chi(p)p + C_X(1 - \chi(p))p = p(C_Y\chi(p) + C_X).$$

$$(7)$$

The conditional first moment of the loss distribution in Eqn. (7) can therefore be used to compare how well the opposing dependence models represent the size of the joint losses, rather than just their conditional probability of occurrence, since the expression includes $C_X$ and $C_Y$. Here, $C_X = C_Y = 1$, hence the conditional expectation of joint loss is equivalent to the conditional expectation of loss jointly occurring at both locations given a loss has occurred at one location. It should be noted that the (non-conditional) expected loss, $\mathbb{E}(L(X,Y))$, does not depend on $\chi(p)$. This is because the expectation of a sum is the sum of the expectations, hence expected total loss over two or more locations is simply the sum of the expected losses at each location, and so is unaffected by the amount of dependency between sites.

Figure 8 presents a comparison of the distribution of the conditional expected joint loss for London paired with each land location in our European domain, given a loss has occurred in London, when calculated empirically and using the two opposing dependence

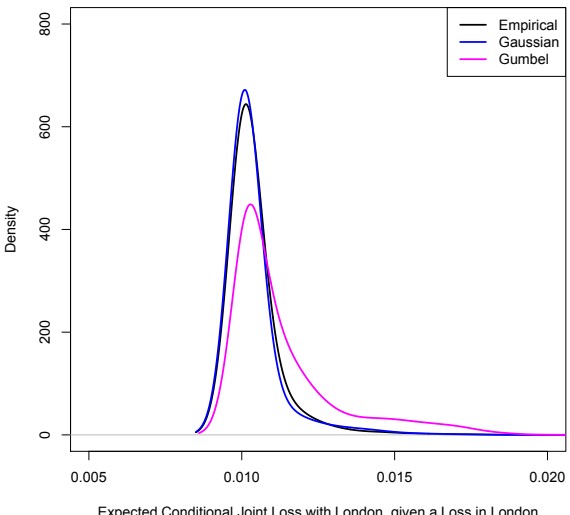

Figure 8: For all land locations in the European domain, the conditional expected joint loss with London, given a loss has occurred in London (Eqn. 7), calculated empirically and using the Gaussian and Gumbel copula models.

models.

Figure 8, further illustrates the importance of correctly specifying extremal dependence class when representing loss. When a conceptual loss occurs in London, the Gumbel dependence model over estimates the expected conditional joint loss with other European land locations, while conversely, the Gaussian model provides a very good estimate of the empirical expected conditional joint loss distribution.

## 4.1 Extending to high dimensions

We extend the analysis in Fig. 7 to systematically explore the high-dimensional domain by fitting both the Gaussian and Gumbel models to a stratified sample of 100 locations paired with each of the other 99 locations, and, for each pair, plot the difference between empirical and modelled $\chi(0.01)$ against their separation distance, shown in Fig. 9.

This domain-wide comparison indicates that, while the Gaussian model slightly over and under estimates empirical $\chi(0.01)$ at small separation distances, this model consistently outperforms the Gumbel model which overestimates $\chi(0.01)$ for all separation distance, even very small. This indicates, as in Fig. 6, that a majority of nearby locations do not exhibit asymptotic dependence as they are not well represented by the

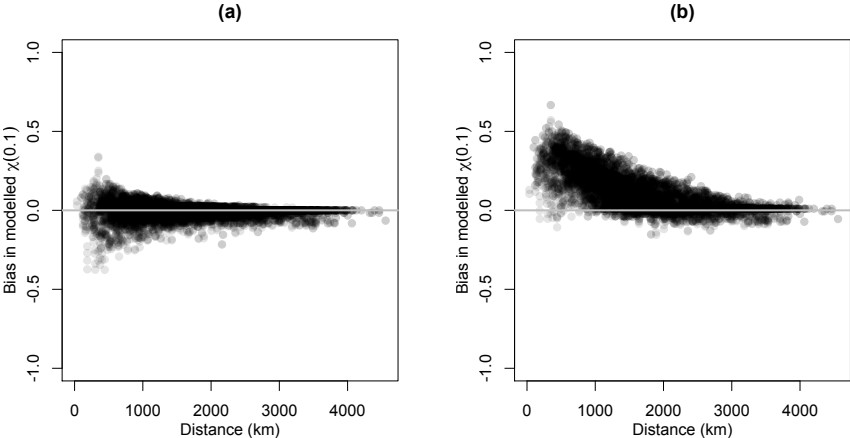

Figure 9: The difference between empirical and modelled $\chi(0.01)$ for a stratified sample of 100 locations paired with each of the other 99 locations, plotted against separation distance for (a) the Gaussian model and (b) the Gumbel model.

Gumbel model, further supporting the diagnosed dominance of extremal independence throughout the European domain.

Finally, we extend the analysis in Fig. 8 to systematically explore the high-dimensional domain by replacing London as the location of origin, with each location within a stratified sample of 100 locations. For each of these 100 locations, Fig. 10 presents the the difference between modelled and empirical relative frequencies of binned ranges of conditional expected joint loss, separately for the Gaussian and Gumbel models, i.e. representing the difference between the modelled and empirical density plots in Fig. 8, but for 100 locations rather than one. Fig. 10 (b) identifies that the discrepancy between the empirical and Gumbel estimates of conditional expected joint loss shown in Fig. 8 are consistent throughout the domain, with lower values being under-represented and higher values over-represented by the Gumbel model. In a similar way, Fig. 10 (a) shows that the Gaussian model performs equally well for these 100 locations, with much smaller discrepancy compared to the Gumbel model, as found in Fig. 8.

This novel conceptual aggregate loss diagnostic approach supports the use of the Gaussian model when asymptotic independence is found to be the dominant extremal dependence characteristic within a high dimensional natural hazards dataset. In this windstorm footprint application, we found that while the Gumbel model is able to represent some pairs of locations at very small separation distances, where asymptotic depen-

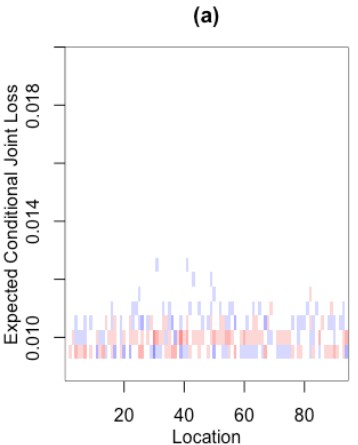
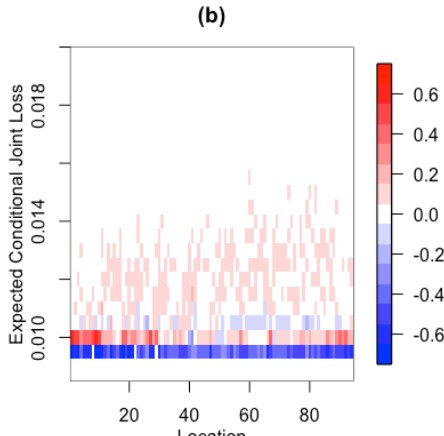

Figure 10: For a stratified sample of 100 locations within the windstorm footprint domain, the difference between modelled and empirical relative frequencies of binned ranges of expected conditional joint loss, for (a) the Gaussian model, (b) the Gumbel model.

dence is suggested by the coefficient of tail dependence, this model greatly misrepresents the joint tail behaviour and hence the conditional probability of joint loss for a majority of pairs and separation distances. Conversely, the Gaussian model is able to represent the joint tail behaviour relevant for loss estimation for locations within close proximity to each other, as well as further apart.

As previously mentioned, alternative windstorm loss thresholds have been implemented in other studies, for example the 98% quantile in Klawa and Ulbrich (2003), and the fixed thresholds of 20ms$^{-1}$ in Bonazzi et al. (2012) and Dawkins et al. (2016) and 25ms$^{-1}$ in Lamb and Frydendahl (1991) and Roberts et al. (2014). An exploration of the effect of the choice of loss threshold and, indeed loss function, on how the opposing dependence models represent joint losses would be an extremely interesting area of further investigation, however beyond the scope of this study. Dawkins (2016) goes some way in addressing this by presenting a comparison for the 98% quantile and 25ms$^{-1}$ fixed loss thresholds in the same form as Fig. 7. Dawkins (2016) found that the overall suitability of the opposing models remained the same for both threshold, although the discrepancy of the Gumbel model was slightly smaller for the lower, 98% quantile, loss threshold. This was thought to be because modelled exceedances further from the upper limit of the joint distribution were less sensitive to a mis-specification of the extremal dependence characteristic in the Gumbel model.

# 5   Why are wind gust speeds asymptotically independent?

It is of interest to ask whether there might be fundamental fluid dynamical reasons for why extreme wind gust speeds should be asymptotically independent at different spatial locations. One approach to answering this question is to consider extremal dependence in turbulent flows. The atmospheric flow in storm track regions is highly chaotic and irregular and is therefore turbulent rather than smoothly varying laminar flow (see Held 1999; and references therein). Furthermore, over short enough spatial distances, the horizontal flow in a storm may be considered to be stationary in space and directionally invariant, in other words, homogeneous isotropic turbulence.

It is useful to first consider the more tractable problem of dependency in simultaneous wind speeds rather than maximum wind speeds over a given time period. The dependency between maximum gust speeds over 3 days will not generally be less than the dependency between simultaneous wind gust speeds because maximum wind gusts for a storm do not occur at the same time at different locations. However, for locations that are close to one another, maximum gust speeds for fast moving extreme storms will occur within a short time window (e.g. within around 3 hours or less for extreme storms over the UK) and so simultaneous results become more relevant.

As originally proposed by Von Kármán (1937), turbulent wind fields can be efficiently and realistically simulated using stochastic processes (Mann, 1998). This approach is widely used for many applications such as testing loads on new aircraft designs. The basic assumption in homogeneous turbulence is that the Cartesian velocity components are independent Gaussian processes, each with a prescribed spatial covariance function. In the special case of isotropic turbulence, the spatial covariance functions are identical for each velocity component. Hence, for 2-dimensional windstorm gusts, the wind gust speed at spatial location, $s$, is given by $X(s) = \sqrt{u^2 + v^2}$, where $u = u(s)$ and $v = v(s)$ are independent Gaussian processes having identical covariance functions.

The distribution of each velocity component is expected, by the Central Limit Theorem, to be close to normally distributed since the net displacement of a particle in turbulence is the result of many irregular smaller displacements. The distribution of each component has zero skewness due to the symmetry of the fluid equations (negative

deviations are as likely as positive ones) but can have slightly more kurtosis (i.e. fatter tails) than the normal distribution due to intermittency in the flow. Measurements of velocity components in the atmospheric surface layer reveal that the distributions are near to Gaussian (e.g. Chu et al. (1996)).

So what can be deduced about the extremal dependence class of wind speeds from such turbulence models? Firstly, as shown in Example 5.32 of McNeil et al. (2005), since the individual velocity components are bivariate normal, the individual velocity components are asymptotically independent at different locations e.g. $u_1 = u(s_1)$ and $u_2 = u(s_2)$ are asymptotically independent when $s_1$ differs from $s_2$, and likewise for $v(s)$. Furthermore, it can be shown that the square of each velocity component is also asymptotically independent (see Appendix).

The squared wind speeds at pairs of locations are sums of two such independent components, $(X^2, Y^2) = (u_1^2 + v_1^2, u_2^2 + v_2^2)$, and so it would be counter intuitive if somehow these sums were not also asymptotically independent. Unfortunately a proof of asymptotic independence between $(X^2, Y^2)$ (and hence $(X, Y)$) remains elusive. Nevertheless, the conjecture can be explored using numerical simulation.

By simulating velocities from bivariate normal distributions, we have found no evidence of extremal dependence in wind speeds even when each velocity component is highly correlated. Figure 11 shows an example obtained by simulating a million wind speeds at two locations where the $u$ and $v$ velocity components are independent standard normal variates each with correlation of 0.9 between locations (i.e. the correlation between $u_1$ and $u_2$ is 0.9). The squared wind speeds at each location are chi-squared distributed with 2 degrees of freedom but are not independent: there is positive association clearly visible in Fig. 11(a). To assess extremal dependence, Fig. 11(b) shows how the joint exceedance probability, $\Pr(X^2 > t^2, Y^2 > t^2)$, and the marginal exceedance probability, $\Pr(X^2 > t^2) = \Pr(Y^2 > t^2)$, behave as threshold $t^2$ is varied. As the threshold is increased the joint probability drops to zero faster than the marginal exceedance probability (the curve in Fig. 11(b) is steeper than the dashed line), which suggests that the ratio, the conditional probability of exceedance, equivalent to $\chi$ in Eqn. (1), will tend to zero in the asymptotic limit.

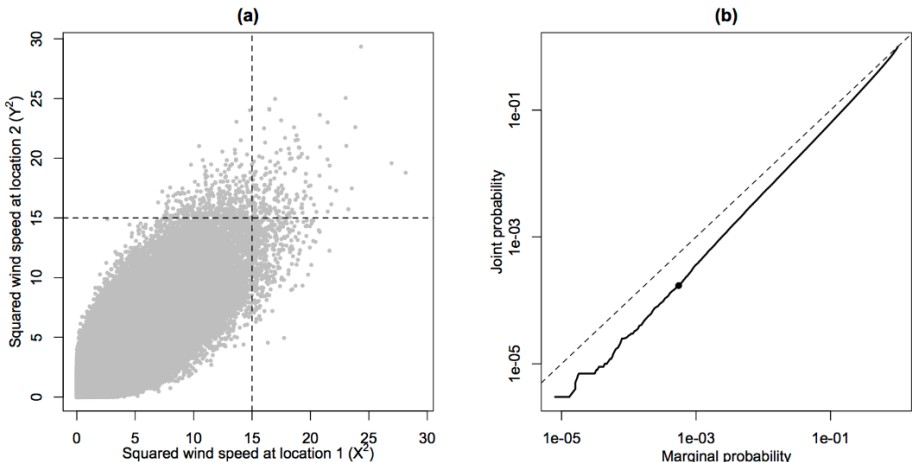

Figure 11: Simulation of wind speeds at two sites having highly correlated velocities (see main text for details): (a) scatter plot of squared wind speeds at the two sites (1000 points randomly sampled out of the million); (b) joint versus marginal exceedance probabilities (on logarithmic axes). The dot shows an example obtained by counting the fraction of points in the upper right and the right hand quadrants of (a). The curve has a steeper slope than the dashed line (equal probabilities denoting complete dependence) suggesting asymptotic independence.

# 6    Conclusion

This study has presented an approach for using the extremal dependence diagnostics of Coles et al. (1999) and Ledford and Tawn (1996) along the the Gaussian and Gumbel copula models to systematically explore the dominant extremal dependence class in a high dimensional natural hazards field. Within this analysis we contribute an additional, natural hazards relevant, aggregate conceptual loss extremal dependence diagnostic approach, again applied to explore extremal dependence in high dimensional spatial data. We find that when a combination of asymptotic independence and dependence is identified within the domain, this aggregate loss diagnostic is beneficial in understanding how using a model for one form of extremal dependence, necessary due to the high dimensionality of the data, effects the representation of this important natural hazards model output, hence providing further justification of the selected dependence model.

These methods reveal strong evidence of the dominance of asymptotic independence in windstorm footprint hazard fields, contrary to what has been assumed in previous studies such as Bonazzi et al. (2012), and that the mis-specification of this extremal dependency

(e.g. by using a Gumbel copula) leads to severe over-estimation of the probability of joint losses. A reason for this lack of asymptotic dependency has been proposed based on arguments from turbulence theory. These results provide justification that spatial representation and simulation of windstorm hazard fields can be represented by a Gaussian geostatistical model, such as that developed in Chapter 5 of Dawkins (2016).

# Acknowledgements

Laura C. Dawkins was supported by the Natural Environment Research Council (Consortium on Risk in the Environment: Diagnostics, Integration, Benchmarking, Learning and Elicitation (CREDIBLE project); NE/J017043/1).

# Appendix

Table 1: Empirical and Parametric forms for extremal dependence measures $\chi(p)$ and $\bar{\chi}(p)$.

| | $\chi(p)$ | $\bar{\chi}(p)$ |
|---|---|---|
| **Empirical** | $\frac{a}{a+c}$ | $\frac{2\log(a+c)/n}{\log(a/n)} - 1$ |
| **Power Law** | $\frac{1}{n}\exp\left(\frac{\alpha}{\eta}\right)p^{\frac{1}{\eta}-1}$ | $\frac{2\log(p)}{\log\left(\frac{1}{n}\exp\left(\frac{\alpha}{\eta}\right)\right)+\frac{1}{\eta}\log(p)} - 1$ |
| **Gumbel** | $\sim 2 - \frac{(2\log(1-p)^r)^{\frac{1}{r}}}{\log(1-p)} = 2 - 2^{\frac{1}{r}}$ (Coles et al., 1999) | $\frac{2\log(p)}{\log(2p(1-p)^2)} - 1$ |
| **Gaussian** | $\bar{F}(1-p, 1-p)/p,$ <br><br> where $\bar{F}(1-p, 1-p) = Pr(X > x_{1-p}, Y > y_{1-p}) \sim (1+\rho)^{\frac{3}{2}}(1-\rho)^{\frac{1}{2}}(4\pi)^{-\frac{\rho}{1+\rho}}(-\log(p))^{\frac{\rho}{1+\rho}}p^{\frac{2}{1+\rho}}$ as $p \to 0$ (Coles et al., 1999) | $\frac{2\log(p)}{\log(\bar{F}(1-p,1-p))} - 1$ |

# Proof of independence in stochastic models of turbulent flows

Assume the velocity components $(u_1, v_1)$ and $(u_2, v_2)$ at two separate locations in an isotropic turbulent flow can be represented as bivariate normally distributed vectors $(u_1, u_2)$ and $(v_1, v_2)$ that are independent and identically distributed with zero expectations.

The individual velocity components, $(u_1, u_2)$ and $(v_1, v_2)$, are both asymptotically independent because of each being bivariate normally distributed.

The squares of the individual velocity components, e.g. $(u_1^2, u_2^2)$, are also asymptotically independent. This is proven by rewriting the joint probability of exceedance:

$$\Pr(u_1^2 > t^2, u_2^2 > t^2)$$

$$= \Pr(u_1 > t, u_2 > t) + \Pr(u_1 > t, u_2 \leq t) + \Pr(u_1 \leq t, u_2 > t) + \Pr(u_1 \leq t, u_2 \leq t)$$

$$= \chi_{++}\Pr(u_1 > t) + \chi_{-+}\Pr(u_1 > t) + \chi_{-+}\Pr(u_1 \leq t) + \chi_{--}\Pr(u_1 \leq t)$$

$$= \chi_{++}\Pr(u_1^2 > t^2) + \chi_{+-}\Pr(u_1^2 > t^2),$$

which is obtained by noting that $\Pr(u_1^2 > t^2) = \Pr(u_1 > t) + \Pr(u_1 \leq t)$, and conditional probabilities $\chi_{++} = \chi_{--}$ and $\chi_{+-} = \chi_{-+}$ by symmetry of the bivariate normal distribution about $(0,0)$. Since the components are bivariate normal, $\chi_{++}$ and $\chi_{+-} \to 0$ as $t \to \infty$, and so $\Pr(u_1^2 > t^2, u_2^2 > t^2)/\Pr(u_1^2 > t^2) \to 0$. Hence, the square of the velocity component is also asymptotically independent.

Perhaps rather counter-intuitively, the sum of two independent identically distributed asymptotically independent variables is not necessarily asymptotically independent. It, therefore, remains to be proven whether or not $(u_1^2 + v_1^2, u_2^2 + v_2^2)$ is asymptotically independent.

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
