# Peer review of "Quantification of extremal dependence in spatial natural hazard"

_Natural Hazards and Earth System Sciences, 2018_

## Referee Comment (RC1) · Anonymous Referee #1 · 9 May 2018

General comments

The aim of this paper is to examine the extremal dependence of windstorm footprint data for pairs of sites across Europe - to assess whether there a non-zero probability that there will be an unusually large wind speed at one site given that such an event has occurred at a second site? Understanding extremal dependence is vital for accurate prediction of joint loss functions for the sites in question. Joint losses are themselves key for natural disasters which affect multiple locations across a large region, in which case the combined loss over this region is of most importance. Traditionally, statistical models for extremal dependence make one of two assumptions: either the data

are asymptotically dependent (eg. max-stable models) or they are asymptotically independent. If the wrong class of extremal dependence is assumed and a model is subsequently fitted then the model predictions cannot be trusted; for example, if it is incorrectly assumed that the largest values at the two sites occur concurrently with non-zero probability then the joint loss at the two sites may be over-estimated.

In this paper, a variety of model-based estimates of the extremal dependence for wind speeds are compared to the equivalent empirical estimate and a copula-based method based on the Gaussian copula if found to give the best fit, although this data-based justification is illustrated by only a small number of sites (London-Amsterdam and London-Madrid). A justification based on a physical model for turbulent wind-speeds is also postulated. Finally the impact of the choice of extremal dependence class on the estimate of a particular definition of the conditional loss function is illustrated.

The paper is mostly well-written, although there are a number of grammatical/typographical errors which should be addressed (see below). The logical flow is clear, and results are well labelled and clearly discussed in the text.

Specific comments

— Comparison of Gumbel, Gaussian and power law copulas to the empirical estimate is shown only for a very few pairs of sites. At the end of Section 3.3 it is suggested that the results found at these sites are representative of results for other pairs of locations. How many other pairs of locations were tested? How confident are you that asymptotic independence is the dominant dependence structure across all pairs of sites? Have you considered ways in which you could formally test this over all pairs of sites?

— A thought on the justification for asymptotic independence based on the model for wind gust speeds. It is assumed in the physical model that each of the Cartesian

components of wind gusts speeds $U(s)$ and $V(s)$ follows an independent Gaussian spatial process. By properties of the multivariate normal distribution, each the vector of each component $\mathbf{U} = (U(s_1), U(S_2))$ and $\mathbf{V} = (V(s_1), V(S_2))$ at any two sites $s_1$ and $s_2$ follows a bivariate normal distribution and consequently the components of each of $\mathbf{U}$ and $\mathbf{V}$ are asymptotically independent. However the Gaussian process assumption is just another modelling assumption, so the question is how accurate is it, ie. how strong is the evidence in favour of the Gaussian assumption? If actually the speed components followed some other process that had asymptotically dependence bivariate margins then the conclusion would be very different. On a related point, it is not entirely obvious where the equation for $\chi_{max}$ and the expression for $\Pr[u_1^2 > t, u_2^2 > t]$ comes from. Although these expressions are correct, Would it be possible to put a derivation in an appendix?

— The second half of Section 4 (p21 onwards) needs some re-working to clarify the points that are being made. For example, it is not clear why we might be interested in the first and second moments derived after line 326; also these are not al moments but expectation (first moment), conditional expectation (conditional first moment) and variance (function of second and first moment). Why does the expected loss not depend on the extremal dependence between the two sites (line 327)? Could you clarify what is being illustrated (line 331) in this final part of the section? Although the Gumbel/Gaussian discrepancy is clear in Figure 7, it might be informative to also look at spatial plots of the *differences* between the empirical and model-based estimates of each of $\chi(u)$ and $\bar\chi(u)$ to see if there is any spatial clustering in these differences, ie. do the models represent the empirical behaviour better for some regions/distances/directions than others? Finally, how does Figure 8 change with the choice of $p$, and does your model enable you to look at the sizes of losses at the two sites rather than just the probability of a loss jointly occurring at each site?

[Figure]

Technical corrections

— Throughout apposite/apposing should be opposite/opposing.

— Lines 71-76: these sentences are not entirely clear. On line 73 please clarify what the 'parametric representations' are representations of. May also be clearer to split the sentence on lines 73–76 into two sentences, the first to discuss what will be done (ie. two copula dependence models fitted) and the second to explain why there two copulas were chosen.

— line 94: please could you clarify exactly which 'statistical property' is meant here. Something like 'the extremal dependence class estimated from the data'.

— line 135: based *on*

— line 137/138: doesn't quite get across the message that sites separated by different distances/directions will have different levels of dependence. Also, why is this likely to be the case? Have you looked at dependence as a function of distance/direction?

— line 165: RHS sign in the inequality should be reversed.

— line 167: could append this paragraph to the previous one.

— line 171: no need to state '*empirical* exploration' as it is made clear later in the sentence that the estimate is an empirical one.

— line 177-180: think this sentence can be removed as it doesn't quite fit here and is better covered in Section 3.3

— line 182: clarify that rarity of extreme events in historical data is not specific to this particular data set.

— line 186: not sure that 'model the asymptotic limit' is quite right, maybe 'predict' instead of 'model'. The model can only reflect the (sub-asymptotic) evidence in the data and the only extra information used in obtaining an estimate of the asymptotic limit is the assumption that the model fitted to sub-asymptotic data can be extrapolated to make predictions on higher (asymptotic) levels.

— Notation: switching from $X$ and $Y$ to $Z_1$ and $Z_2$ mixes two different ways of distinguishing sites (different letters *v.* subscripts). Could change $(X, Y)$ to $(X_1, X_2)$ or $(Y_1, Y_2)$, *or* change $(Z_1, Z_2)$ to $(Z_X, Z_Y)$.

— line 196: 'asymptotic' $\rightarrow$ 'extremal', as asymptotic dependence is a particular class of extremal dependence.

— line 198: expression for $\bar{\chi}(0)$ has an excess bracket.

— line 201: 'model' missing after Ledford and Tawn (1996)

— lines 218-220: this sentence would be clearer split into two. First describing the models for $(X, Y)$ and then describing how model-based predictions of $\chi(p)$ and $\bar{\chi}(p)$ are obtained from the models and are compared to the empirical estimate in Figure 3.

— Figure 6 caption $ms^{-1}$.

---

## Referee Comment (RC2) · Anonymous Referee #2 · 16 May 2018

General Comments

The authors explore the joint spatial structure of extreme windstorms and the investigation is of good quality and described very clearly in the manuscript. I ask for a revision to address some outstanding issues, before publication.

Specific Comments

1. Observational errors

The analysis is performed on mean winds from a numerical model which are post-processed to gusts using a highly simplified model given on Line 124. These estimates

of gusts will differ considerably from the true storm-max gusts experienced at sites and the influence of this error on results has potential to be significant. Therefore, errors in estimated gusts need to be quantified, and their impacts on results should be measured and presented to readers.

A comparison of MetUM-derived estimates with weather station data would provide a realistic measure of observational uncertainty. I suggest the rms difference in max storm gust between the authors' dataset (grid-cell encompassing Heathrow) and observed gusts for Heathrow is computed using the top N storm max gusts *observed* at Heathrow, where N is approx. 50 to focus on tail extremes. GSOD is a free source of observed weather for many stations, including Heathrow.

The confidence intervals in Figure 4 of original manuscript are based on sampling error and need replaced to include these estimates of observational error for each MetUM storm max gust. Figures 3 and 5b would also benefit from the inclusion of estimates of uncertainty in plotted values, due to both observational and sampling errors.

2. Events analysed in Figure 4, and interpretation of results

Fig 2 indicates approx. 25 points above quantile=0.99, which suggests that the quantile=0.5 in Figure 4 is based on over one thousand storm events in a 35 year period. The inclusion of about 30 events per year on average will contain many breezy days. These data points are potentially misleading to include, because the spatial structure of days with weak winds is likely to be substantially different from the spatial structure of severe events producing tail winds. I request that Figure 4 is re-drawn using data from quantile=0.9 and upwards. This would still include weak winds from minor cyclones, but is a step in the right direction, while maintaining sufficient sample sizes.

The conclusions to be drawn from Figure 4a should be reviewed in a revised version of manuscript. First, the results in Figure 4a indicate rising values of the coefficient of tail dependence for quantile thresholds above 0.9, towards a value of unity for the highest quantile. Given the aim to capture behaviour in the limit as p tends to 0, it seems unsafe

to conclude that London-Amsterdam has tail independence. Second, the inclusion of observational uncertainty (point 1 above) will broaden the confidence limits which may require a new interpretation of results.

3. Section 3.4 conjecture

The conjecture to explain the tail independence in section 3.4 begins by representing storm winds as isotropic turbulence. It is standard to represent storm winds as the sum of a mean wind and a smaller turbulent contribution. This is also consistent with the MetUM model gust dataset used by the authors (description around Line 124).

The authors then assume that gusts at two locations are bi-variate normal. While the turbulent contribution to winds at two distinct locations might be bi-variate normal at any instant in time, the gusts analysed for tail dependence are the maximum gust over the whole storm. The storm-max gusts between neighbouring locations are expected to have strong tail dependence since they would have very similar mean wind and max gustiness from isotropic turbulence.

Regarding the text on Lines 254-258: McNeil et al. (2005) showed that if correlation is less than one, then the coefficient of upper tail dependence equals zero, their Example 5.32.

McNeil, A J, Frey, R, and Embrechts, P: Quantitative Risk Management Concepts, Techniques, Tools. Princeton University Press, 2005

4. Section 4 on losses

Lines 277-283: the authors state a simple loss function provides better storm loss estimates than the Klawa and Ulbrich (KU) loss function. There are various reasons why this judgement on loss functions is misleading. Besides the minor fact that the two articles quoted excluded population weighting hence did not test the KU loss function, there is a more significant issue that 'better' is defined in non-standard and highly specific terms as 'a subset of 23 significant storms completely contained in highest

quantile of all storms'. Further, there is much published work on how loss severity is a function of wind speed, and KU's loss function certainly captures this effect more accurately than a step function.

The 'conceptual loss function' used by the authors could be more accurately described as areal frequency of loss occurrence, and ignoring loss severity. Its usefulness in estimating total loss is an interesting result, since it suggests the area of storm above a loss-causing threshold is the dominant contributor to total storm loss. I suggest the authors describe their loss function in more specific terms as 'areal frequency of loss' in the text.

Further, if the authors wish to retain text comparing a step function to the superior KU loss function, then the authors should include more information for readers: errors in loss estimates depend on wind speeds, loss functions and exposure density, and the success of the simplest loss function over KU in tests performed by Roberts et al. is very likely due to its relative insensitivity to errors in other components of their loss estimates, such as estimated gusts. This helps resolve the dilemma of a rapid growth of loss with windspeed indicating KU, while a less sophisticated testing framework indicated a step function.

The over-estimation of joint loss probabilities in the maps in Figures 7e & f are explained as a mis-specification of asymptotic dependence (lines 308-309). However, it could be due to a too high estimate of the dependence parameter r. Could the authors include in the text the group of data used for estimating the dependence parameter?

Technical Corrections

There are many instances of 'apposing' when 'opposing' may be more appropriate?

---

## Referee Comment (RC3) · Anonymous Referee #3 · 17 May 2018

The study investigates whether the commonly-made assumption of asymptotic extremal dependence in windstorm footprint between geographically remote locations is supported by data and if not, what a suitable alternative approach may be. This is a very important question, and the introduction motivates in a clear and concise manner the relevance of the study. The rest of the paper is also generally well-written and clearly structured. As such, form-wise it would be suitable for publication after some very minor revisions. I have more serious concerns regarding the substance of the study: in particular, the basis for the repeated claims of novelty in the introduction, a seeming lack of contextualization relative to the very broad field of extremal dependence modelling and the fact that the whole analysis is centred on two case studies. I

Printer-friendly version

Discussion paper

detail these and some other minor points below.

**Main Comments**

1. a. The authors repeatedly claim the novelty of their approach (ll. 63, 100, 103). As they implicitly note on ll. 63, the main novelty lies in the combination of existing modelling approaches rather than in some fundamental statistical advance. However, conceptually very similar approaches for investigating the appropriate dependence class for spatially remote geophysical extreme events have been implemented before, within a more comprehensive theoretical framework (for example, see Kereszturi et al. 2016). Other than being applied to a different variable, what broad additional insights does the present study provide?
- b. On a related note, the authors suggest that an important result of their work will be to simplify the development and use of models that correctly represent extremal dependence for the variable of interest, removing the need to apply more complex – but more flexible – models which account for the different possible dependence classes (ll. 91-95). There are a number of these models available, including those of Wadsworth et al. (2017) and Huser and Wadsworth (2018). The actual benefits of the approach proposed by the authors are not explicitly described in the manuscript. Are the authors suggesting that the final result stemming from their approach outperforms these models (or that the results are comparable but require less work?) If so a comparison should be provided. Or that the reduction in computational time is so large as to make a difference in practical applications (if so, some indicative figures should be provided)? Or that the ease of implementation of their approach makes it applicable to datasets where other models couldn't be applied? Again, some examples should be provided and the extent/range of validity of this advantage should be discussed. Any one of the above points would be a sound motivation for the present work, but they would need to be explicitly stated and factually supported.
- c. As a final note, very little is said in the introduction of the above-mentioned models

which account for a broad range of dependency classes (see also references in Huser et al., 2017). There is a growing literature in this subfield, which should be discussed.

With the above I don't suggest that the work of the authors is devoid of interest, but they should certainly explain more clearly what the real novelty of the study and what the advantages it will provide to the community are. In my view, it will not be sufficient to alter one or two sentences in the manuscript: this will require a substantial clarification and contextualization effort, and likely some additional analysis to support the claims made.

2. My second major concern regarding this study is the fact that the results are presented only for two location pairs (with one location common to both). The authors briefly mention the fact that they have tested their results for other locations (ll. 250-252), but this is not substantiated in any meaningful way. Is there a way to systematically test the robustness of the results obtained by the authors across a western European domain, perhaps presenting the results in a form similar to Fig. 7 but for different reference locations or a 2-D version of Fig. 8 showing location on one axis, conditional joint loss on the other and density as colours/contours?

**Additional Comments**

3. The title suggests a very broad relevance of the paper. Even though the techniques discussed in the study are general, the analysis effectively focusses on windstorms at three specific locations. As such, the current title is misleading and should be changed to reflect the contents of the study. Alternatively, the approach proposed by the authors should be applied to other geophysical variables and geographical domains.

4. l. 124: Please include a reference for how the wind gusts are calculated. This parametrisation is very simple. If it works well, simplicity is obviously good, but a brief discussion of its performance versus alternative approaches should be provided.

5. Section 3.4: Are wind gust speeds really independent Gaussian processes? Can

Printer-friendly version

Discussion paper

this be tested on the data available to the authors?

6. Fig. 1. The labels/city names are very difficult to see in print.

**References**

Keresztfuri, M., Tawn, J., & Jonathan, P. (2016). Assessing extremal dependence of North Sea storm severity. *Ocean Engineering*, 118, 242-259.

Huser, R., Opitz, T., & Thibaud, E. (2017). Bridging asymptotic independence and dependence in spatial extremes using gaussian scale mixtures. *Spatial Statistics*, 21, 166-186.

Huser, R. G., & Wadsworth, J. L. (2018). Modeling spatial processes with unknown extremal dependence class. *Journal of the American Statistical Association*.
* * *
Printer-friendly version

Discussion paper

---

## Author Comment (AC1) · 10 Aug 2018

Key:

- **Reviewer's comment**

- *Our response*

- Additional/edited text in the manuscript

[Figure]

**1 Reviewer 1**

**1.1 Specific comments**

- **Comparison of Gumbel, Gaussian and power law copulas to the empirical estimate is shown only for a very few pairs of sites. At the end of Section 3.3 it is suggested that the results found at these sites are representative of results for other pairs of locations. How many other pairs of locations were tested? How confident are you that asymptotic independence is the dominant dependence structure across all pairs of sites? Have you considered ways in which you could formally test this over all pairs of sites?**

  *This is a very good point, and important to demonstrate. We have addressed this within the paper at the end of Section 3.3 by adding a few paragraphs and some additional analysis:*

  When aiming to develop a statistical model for high dimensional spatial data over a large geographical domain, it is essential to systematically explore the dominant extremal dependence class across all locations. Here, we present an approach for doing so, which uses this quick-to-calculate coefficient of tail dependence diagnostic, demonstrated by application to our windstorm footprint data set. We first take a stratified (based on the distribution of locations over longitude and latitude) sample of 100 locations within the European domain. One such sample is shown in [Figure 1 in attachments](a). Since the extremal dependence is likely to decrease with increasing separation distance (Wadsworth and Tawn (2012)) and we hope to understand if asymptotic independence is dominant and hence present at small separation distances, for each of these 100 locations, we estimate the coefficient of tail dependence, $\eta$ (and the associated

95% profile likelihood confidence interval) when paired with the 100 nearest locations within the full domain. [Figure 1 in attachments](b) demonstrates how the 100 nearest locations are geographically distributed for one such sampled location in our windstorm footprint dataset. For each pairing, the coefficient of tail dependence is calculated using w as the 0.9 quantile threshold of the structure variable, found to ensure stable estimates of $\eta$ using diagnostic plots as in [Figure 1 in attachments] (c). The estimated $\eta$ parameters and confidence intervals for these 100×100 pairs of locations are plotted against separation distance to explore how, throughout the domain, $\eta$ varies at small separation distances and changes with increasing separation distance, shown in [Figure 1 in attachments] (d). The parameter estimate related to the pair of locations in pink and blue in [Figure 1 in attachments] (b) is shown in pink. This method is repeated many times with 10 such repetitions shown in [Figure 1 in attachments] of the Supplementary Material at the end of the paper, showing very similar results.

[Figure 1 in attachments here] - (a) A stratified (based on the distribution of locations over longitude and latitude) sample of locations within the European domain, with stratified grid shown in grey; (b) a demonstration of the 100 nearest locations [turquoise] to one of these sampled locations [blue], with one such point selected at random [pink]; (c) the coefficient of tail dependence diagnostic plot (as in Fig. 4) for wind gusts at the blue location paired with the pink location; (d) the coefficient of tail dependence (estimated using w as the 0.9 quantile threshold of the structure variable) and 95% profile likelihood confidence intervals, for each of the 100 sampled locations paired with their 100 nearest locations in the full domain, plotted against separation distance in kilometres, with the estimate based on the pair of locations in (b) and (c) added in pink.

[Figure 1 in attachments] (d) shows that for small separation distances ($<$180 km) a proportion of pairs of locations have coefficients of tail dependence parameter, $\eta$, estimates close to 1, with $\eta = 1$ within the confidence interval, indicating asymptotic dependence. Within the range (0-50 km) 69% of pairs of locations exhibit this behaviour, however this proportion reduces rapidly as separation distance increases, to 30% for locations separated by (50-100 km), 13% for locations separated by (100-150 km) and 3% for locations separated by (150-200 km). Hence, while there is evidence of asymptotic dependence for some locations in close proximity, even at very small separation distances (50 km) a larger proportion of locations exhibit asymptotic independence. Indeed, here and in [Figure 2 in attachments] of the Supplementary Material, beyond a separation distance of approximately 200km the coefficients of tail dependence parameter estimates drop well below 1, indicating asymptotic independence. Therefore, since separation distances within the domain extend to up to 3500km, we conclude that asymptotic independence is the dominant extremal dependence structure across the spatial domain.

It is important to consider the validity of representing even this small proportion of asymptotically dependent pairs of locations incorrectly as asymptotically independent. To explore this, Bortot et al. (2000) carried out a simulation study in which they fit the Gaussian, Ledford and Tawn (1996) and Gumbel models to bivariate data simulated from three parent populations with different classes of extremal dependence. They conclude that, for asymptotically independent parent populations the Gaussian copula is able to provide accurate inferences for tail probability estimates, out performing the Gumbel copula model, and even for asymptotically dependent parent populations, the estimation error of the Gaussian copula model was deemed to be acceptably small. This suggests that, when data dimensionality prohibits the use of flexible extremal dependence

models, such as Huser and Wadsworth (2018), and asymptotic independence is found to be the dominant extremal dependence structure across the spatial domain, using an asymptotically independent model, such as the Gaussian tail model, is preferable over using a model for asymptotic dependence throughout the domain. In Section 4 we present a further, natural hazards relevant, diagnostic approach for further validating this, based on an estimates of the aggregate natural hazard losses.

[Figure 2 in attachments here] - For 10 stratified samples of 100 locations within the European domain: the coefficient of tail dependence (estimated using w as the 0.9 quantile threshold of the structure variable) and $95\%$ profile likelihood confidence intervals, for each of the 100 sampled locations paired with their 100 nearest locations in the full domain, plotted against separation distance in kilometres.

- **A thought on the justification for asymptotic independence based on the model for wind gust speeds. It is assumed in the physical model that each of the Cartesian components of wind gusts speeds $U(s)$ and $V(s)$ follows an independent Gaussian spatial process. By properties of the multivariate normal distribution, each the vector of each component $U = (U(s1), U(S2))$ and $V = (V(s1), V(S2))$ at any two sites $s1$ and $s2$ follows a bivariate normal distribution and consequently the components of each of U and V are asymptotically independent. However the Gaussian process assumption is just another modelling assumption, so the question is how accurate is it, ie. how strong is the evidence in favour of the Gaussian assumption? If actually the speed components followed some other process that had**

**asymptotically dependence bivariate margins then the conclusion would be very different.**

*The reviewer raises a good point. This brief justification has been added to the article in the third paragraph of Section 3.4:*

The distribution of each velocity component is expected by the Central Limit Theorem to be close to normally distributed since the net displacement of a particle in turbulence is the result of many irregular smaller displacements. The distribution of each component has zero skewness due to the symmetry of the fluid equations (negative deviations are as likely as positive ones) but can have slightly more kurtosis (i.e. fatter tails) than the normal distribution due to intermittency in the flow. Measurements of velocity components in the atmospheric surface layer reveal that the distributions are near to Gaussian (e.g. Chu et al. (1996)).

- **On a related point, it is not entirely obvious where the equation for $\chi_{max}$ and the expression for $\mathbf{Pr}(u_1^2 > t, u_2^2 > t)$ comes from. Although these expressions are correct, would it be possible to put a derivation in the appendix?**

*It's reassuring that the expressions are correct and the sensible suggestion of adding a short appendix has been adopted:*

Assume the velocity components $(u_1, v_1)$ and $(u_2, v_2)$ at two separate locations in an isotropic turbulent flow can be represented as bivariate normally distributed vectors $(u_1, u_2)$ and $(v_1, v_2)$ that are independent and identically

distributed with zero expectations.

The individual velocity components, $(u_1, u_2)$ and $(v_1, v_2)$, are both asymptotically independent because of each being bivariate normally distributed.

The squares of the individual velocity components, e.g. $(u_1^2, u_2^2)$, are also asymptotically independent. This is proven by rewriting the joint probability of exceedance:

$$\Pr(u_1^2 > t^2, u_2^2 > t^2)$$
$$= \Pr(u_1 > t, u_2 > t) + \Pr(u_1 > t, u_2 \le t) + \Pr(u_1 \le t, u_2 > t) + \Pr(u_1 \le t, u_2 \le t)$$
$$= \chi_{++}\Pr(u_1 > t) + \chi_{-+}\Pr(u_1 > t) + \chi_{-+}\Pr(u_1 \le t) + \chi_{--}\Pr(u_1 \le t)$$
$$= \chi_{++}\Pr(u_1^2 > t^2) + \chi_{+-}\Pr(u_1^2 > t^2),$$

which is obtained by noting that $\Pr(u_1^2 > t^2) = \Pr(u_1 > t) + \Pr(u_1 \le t)$, and conditional probabilities $\chi_{++} = \chi_{--}$ and $\chi_{+-} = \chi_{-+}$ by symmetry of the bivariate normal distribution about $(0, 0)$. Since the components are bivariate normal, $\chi_{++}$ and $\chi_{+-} \to 0$ as $t \to \infty$, and so $\Pr(u_1^2 > t^2, u_2^2 > t^2)/\Pr(u_1^2 > t^2) \to 0$. Hence, the square of the velocity component is also asymptotically independent.

Perhaps rather counter-intuitively, the sum of two independent identically distributed asymptotically independent variables is not necessarily asymptotically independent. It, therefore, remains to be proven whether or not $(u_1^2 + v_1^2, u_2^2 + v_2^2)$ is asymptotically independent.

- **The second half of Section 4 (p21 onwards) needs some re-working to clarify the points that are being made. For example, it is not clear why we might be interested in the first and second moments derived after line 326; also these are not all moments but expectation (first moment), conditional expectation (conditional first moment) and variance (function of second and first moment).**

*This part of Section 4 has now been rewritten more clearly present our motivation and correctly refer to these quantities. In combination with your comment below - **does your model enable you to look at the sizes of losses at the two sites rather than just the probability of a loss jointly occurring at each site?**, we have included a more detailed review of alternative windstorm loss functions, in which the size of the loss is represented as a function of the wind (rather than being equal to 1 as it is in our loss function). We have changed the definition of the conceptual loss function to a more generic form which quantifies the size of the losses as well as the exceedance of the loss threshold.*

*Within this section we initially present the probability mass function of this generic bivariate conceptual loss function, demonstrating how the success of a given model in representing the bivariate conceptual loss for the pair $(X, Y)$ closely relates the its characterisation of $\chi(p)$, and hence present a comparison of empirical and modelled $\chi(0.01)$ and $\bar{\chi}(0.01)$ for the 2 opposing dependence models (Gaussian/Gumbel).*

*We then derive the conditional expected loss for the generic loss function which is a function of $\chi(p)$ as well as the the size of the loss at each location, therefore motivating the use of this conditional first moment for comparing how well the Gaussian and Gumbel models represent the size of the joint losses, rather than just their conditional probability of occurrence.*

*After paragraph 1 in Section 4 we have edited:*

[revised manuscript text omitted]

*Followed finally by Fig. 8*

- **Why does the expected loss not depend on the extremal dependence between the two sites (line 327)?**

*We agree this need clarification. We have added this explanation to the end of the final paragraph in the previous response:*

It should be noted that the (non-conditional) expected loss, $\mathbb{E}(L(X, Y))$, does not depend on $\chi(p)$. This is because the expectation of a sum is the sum of the expectations, hence expected total loss over two or more locations is simply the sum of the expected losses at each location, and so is unaffected by the amount of dependency between sites.

- **Could you clarify what is being illustrated (line 331) in this final part of the section?**

*This sentence has been removed in the rewriting of Section 4.*

- **Although the Gumbel/Gaussian discrepancy is clear in Figure 7, it might be informative to also look at spatial plots of the differences between the empirical and model-based estimates of each of chi(u) and chibar(u) to see if there is any spatial clustering in these differences, ie. do the models repre-**

**sent the empirical behaviour better for some regions/distances/directions than others?**

*We have now change Figure 7 to show the difference between the empirical and modelled estimates of $\chi$ and $\bar{\chi}$ and changed the interpretation of the plot accordingly.*

[Figure 3 in attachments here] - The difference between empirical and modelled $\chi(0.01)$ for (a) the Gaussian model and (b) the Gumbel model, and the difference between empirical and modelled $\bar{\chi}(0.01))$ for (c) the Gaussian model and (d) the Gumbel model, for London paired with all other locations over land.

[Figure 3 in attachments] demonstrates how the Gaussian model is able to represent empirical $\chi(0.01)$ well throughout the domain. Conversely the Gumbel model greatly over estimates $\chi(0.01)$ for all pairs of locations with non-zero empirical $\chi(0.01)$, bar the neighbouring grid cell. However, this neighbouring location is also well represented by the Gaussian model. The Gaussian model reproduces $\bar{\chi}(0.01)$ well for locations within a small to medium separation distance from London, with this distance being greater in the West-East direction, reflecting the common path of storms over Europe (Hoskins and Hodges (2002)). The Gaussian model over and under estimates $\bar{\chi}(0.01)$ for far away locations, with underestimation particularly in furthest away locations. This discrepancy is most likely due to the very small sample of joint extremes at these pairs of locations making estimates of $\bar{\chi}(0.01)$ highly uncertain. The Gumbel model greatly overestimates $\bar{\chi}(0.01)$, for all location, except again for those locations in very close proximity to London. This discrepancy in the Gumbel model is likely

due to a misspecification of asymptotic dependence between most locations, resulting in an overestimation of the conditional dependencies in the extremes.

- **Finally, how does Figure 8 change with the choice of p?**

*While this would be interesting to explore we feel that this is beyond the scope of the study. We have chosen to present the results for one conceptual loss function with the selection of this function and the the value of p justified in the text. I (Dawkins) have, however, addressed this in part in previous work published in my PhD thesis (Dawkins et al. (2016)), identifying no change in the overall results when $p$ is varied. We have now added a few sentences to the end of Section 4 to acknowledge this point:*

As previously mentioned, alternative windstorm loss thresholds have been implemented in other studies, for example the $98\%$ quantile in Klawa and Ulbrich (2003), and the fixed thresholds of 20ms$^{-1}$ in Bonazzi et al. (2012) and Dawkins et al. (2016) and 25ms$^{-1}$ in Lamb and Frydendahl (1991) and Roberts et al. (2014). An exploration of the effect of the choice of loss threshold and, indeed loss function, on how the opposing dependence models represent joint losses would be an extremely interesting area of further investigation, however beyond the scope of this study. Dawkins (2016) goes some way in addressing this by presenting a comparison for the $98\%$ quantile and 25ms$^{-1}$ fixed loss thresholds in the same form as [Figure 3 in attachments]. Dawkins (2016) found that the overall suitability of the opposing models remained the same for both threshold, although the discrepancy of the Gumbel model was slightly smaller for the lower, $98\%$ quantile, loss threshold. This was thought to be because modelled exceedances further from the upper limit of the joint distribution were less

sensitive to a misspecification of the extremal dependence characteristic in the Gumbel model.

- **And does your model enable you to look at the sizes of losses at the two sites rather than just the probability of a loss jointly occurring at each site?**

*This point has been addressed in the response above in which we restructure Section 4 and explain how the analysis could be extended to explore the size of the losses.*

**1.2  Technical corrections**

- **Throughout apposite/apposing should be opposite/opposing.**

*Thank you we have changed these.*

- **Lines 71-76: these sentences are not entirely clear. On line 73 please clarify what the 'parametric representations' are representations of. May also be clearer to split the sentence on lines 73–76 into two sentences, the first to discuss what will be done (i.e. two copula dependence models fitted) and the second to explain why there two copulas were chosen.**

*Thank you we have changed this to:*

For a given pair of locations within a windstorm hazard field, we fit Gumbel and Gaussian bivariate copula dependence models, and explore how well these models represent the empirical estimates of $\chi(p)$ and $\bar{\chi}(p)$. These two copula models characterise opposing extremal dependence class, the Gaussian

copula characterising asymptotic independence and the Gumbel asymptotic dependence, hence this comparison gives an indication of the form of extremal dependence within the data.

- **line 94: please could you clarify exactly which 'statistical property' is meant here. Something like 'the extremal dependence class estimated from the data'.**

*This has been removed in the restructuring of Section 4*

- **line 135: based on**

*Thank you we have changed this.*

- **line 137/138: doesn't quite get across the message that sites separated by different distances/directions will have different levels of dependence. Also, why is this likely to be the case? Have you looked at dependence as a function of distance/direction?**

*We have previously looked at the correlation in wind gust speeds as a function of distance and have therefore added this to the supplementary material and refer to it at the end of the first paragraph in Section 3:*

These three locations are shown in Fig. 1, and these two pairings are chosen because of their contrasting separation distances, and hence degrees of dependence (as shown in [Figure 4 in attachments] in the Supplementary Material).

[Figure 4 in attachments here] - Empirical correlation between (a) London, (b) Amsterdam and (c) Berlin and all other land locations over land, plotted against distance in (d), (e) and (f) respectively and for distance binned average correlation in (g), (h), (i) respectively.

- **RHS sign in the inequality should be reversed.**

  *Thank you we have now changed this.*

- **line 167: could append this paragraph to the previous one.**

  *Thank you we have now changed this.*

- **line 171: no need to state 'empirical exploration' as it is made clear later in the sentence that the estimate is an empirical one.**

  *Thank you we have now changed this.*

- **line 177-180: think this sentence can be removed as it doesn't quite fit here and is better covered in Section 3.3**

[Figure]

*We agree and have now removed these lines and refer to Fig 3 in Section 3.3, which we now reference in the caption for Figure 3.*

- **line 182: clarify that rarity of extreme events in historical data is not specific to this particular data set.**

  *Yes, this is true. We have now edited the first paragraph of Section 3.3 to:*

  Here, as in all datasets of environmental phenomena, the rarity of very extreme events makes it impossible to empirically quantify the asymptotic limits $\chi(0)$ and $\bar{\chi}(0)$, necessary for extremal dependence class identification.

- **line 186: not sure that 'model the asymptotic limit' is quite right, maybe 'predict' instead of 'model'. The model can only reflect the (sub-asymptotic) evidence in the data and the only extra information used in obtaining an estimate of the asymptotic limit is the assumption that the model fitted to sub-asymptotic data can be extrapolated to make predictions on higher (asymptotic) levels.**

  *We have now altered the first paragraph of Section 3.3 to take this comment into account:*

  To overcome this, Ledford and Tawn (1996) developed a bivariate tail model, able to characterise both classes of extremal dependence, which when fit to a bivariate random variable can be used to predict the asymptotic limit of the

conditional probability measures and hence give an estimate of the class of extremal dependence, based on the sub-asymptotic evidence in the data and the assumption that the model can be extrapolated to asymptotic levels.

- **Notation: switching from X and Y to Z1 and Z2 mixes two different ways of distinguishing sites (different letters v. subscripts). Could change (X; Y ) to (X1;X2) or (Y1; Y2), or change (Z1;Z2) to (ZX;ZY ).**

*Thank you we have now changed this.*

- **line 196: 'asymptotic' - 'extremal', as asymptotic dependence is a particular class of extremal dependence.**

*Thank you we have now changed this.*

- **line 198: expression for (0) has an excess bracket.**

*Thank you we have now changed this.*

- **line 201: 'model' missing after Ledford and Tawn (1996)**

*Thank you we have now changed this.*

- **lines 218-220: this sentence would be clearer split into two. First describing the models for (X; Y ) and then describing how model-based predictions of chi(p) and chi bar (p) are obtained from the models and are compared to the empirical estimate in Figure 3.**

*We agree and have now made this edit:*

In addition, as a comparison, alternative parametric bivariate dependence models, known as the Gaussian and Gumbel copulas, are used to model the pair $(X, Y)$, since each copula characterises an opposing extremal dependence class. Model based predictions of $\chi(p)$ and $\bar{\chi}(p)$ for each copula are included in Fig. 3. The representation of $\chi(p)$ and $\bar{\chi}(p)$ in the limit $p \to 0$ for these opposing models then gives further indication of the extremal dependence class present.

- **Figure 6 caption ms-1.**

*Thank you we have now changed this.*

---

## Author Comment (AC2) · 10 Aug 2018

Key:

- **Reviewer's comment**

- *Our response*

- Additional/edited text in the manuscript

[Figure]

**0.1 Observational errors**

- **The analysis is performed on mean winds from a numerical model which are post-processed to gusts using a highly simplified model given on Line 124. These estimates of gusts will differ considerably from the true storm-max gusts experienced at sites and the influence of this error on results has potential to be significant. Therefore, errors in estimated gusts need to be quantified, and their impacts on results should be measured and presented to readers.**

  **A comparison of MetUM-derived estimates with weather station data would provide a realistic measure of observational uncertainty. I suggest the rms difference in max storm gust between the authors' dataset (grid-cell encompassing Heathrow) and observed gusts for Heathrow is computed using the top N storm max gusts *observed* at Heathrow, where N is approx. 50 to focus on tail extremes. GSOD is a free source of observed weather for many stations, including Heathrow.**

  *Thank you, this is a very valid point that should be discussed and explored. We have addressed this point in Section 2 by first reviewing the rigorous evaluation of the Met UM footprints in Roberts et al. (2014), and presenting your suggested comparison with the GSOD data. The GSOD weather station within our London grid cell is London City, hence we have used this station rather than Heathrow. We have added this to the end of Section 2:*

  Using model generated windstorm footprints for representing historical storms has benefit in terms of spatial and temporal coverage, however these estimated maximum wind-gust speeds will inevitably differ from the those observed at nearby weather stations. For example, as noted by Roberts et al. (2014), several

alternative methods for parameterising wind gust speeds are available (see Sheridan (2011) for a review), which can lead to large differences in estimated gusts (10-20ms$^{-1}$). The validity of simplistic gust parameterisation stated above was evaluated by Roberts et al. (2014), who found an overestimation in the effect of surface roughness at stations greater than $\sim 500$ metre altitude, leading to underestimation of MetUM modelled extreme winds in these locations. In addition, within this thorough evaluation of MetUM windstorm footprints, Roberts et al. (2014) identified a slight underestimation in extreme wind gust speeds greater than $\sim 25\text{ms}^{-1}$. This was found to be due to a number of mechanisms including the underestimation of convective effects and strong pressure gradients, leading to the underdevelopment of fast moving storms (Roberts et al. (2014)).

[Figure 1 in attachments] (a) The relationship between MetUM windstorm footprint wind gust speeds in the London grid cell and the corresponding observed wind-gust speeds at the London City weather station within the Global Summary Of the Day dataset, and (b) the same relationship for the 50 must extreme windstorm events at the London City weather station.

To explore the possible discrepancy in the MetUM windstorm footprint wind gust speed relevant for this study, we extract daily maximum observed wind gust speed recorded at the London City weather station (the station located within the London grid cell used throughout this study) from the Global Summary Of the Day (GSOD) data repository ($https : //data.noaa.gov/dataset/dataset/global - surface - summary - of - the - day - gsod$), and, for each of the 6103 windstorm events in our dataset, find the maximum observed gust in the 3 day period centred on the same date as in the MetUM model generated footprints. A comparison of the observed and MetUM modelled footprint wind gusts in London is presented in [Figure 1 in attachments] (a), indicating a general overestimation

in modelled wind-gust speeds below $25m^{-1}$ and a slight underestimation for wind-gust speeds above $25m^{-1}$, reflecting the findings of Roberts et al. (2014). [Figure 1 in attachments] (b) presents this same relationship for the 50 most extreme events in the observed dataset, highlighting this underestimation of modelled extreme wind-gust speeds. For example, during the windstorm in which the highest observed wind-gust speed of 34.98 $ms^{-1}$ occurs at the London City station, the MetUM model produces a maximum wind-gust speed of just 23.88 $ms^{-1}$. Indeed, the root mean squared difference between the observed and modelled footprint wind-gust speeds for these 50 extreme events is $4.57ms^{-1}$, giving an indication of the model uncertainty in representing extreme windstorm footprint wind-gust speeds.

The discrepancy in model generated wind-gust speeds compared to the observations could lead to differences in results, namely the identification of the extremal dependence class between locations. To explore this possibility we repeat parts of the analysis presented in the following sections using this GSOD data, discussed further in Section 3. It should also be noted, however, that the station data should not be treated as the true state of the world, since a number of factor, such as instrumental inaccuracies, lead to observational error. A full exploration and quantification of the observational uncertainty present in the MetUM model as well at the observations themselves is beyond the scope of this study, however this discrepancy should be kept in mind when interpreting results.

- **The confidence intervals in Figure 4 of original manuscript are based on sampling error and need replaced to include these estimates of observational error for each MetUM storm max gust. Figures 3 and 5b would also benefit from the inclusion of estimates of uncertainty in plotted values, due**

**to both observational and sampling errors.**

*While this would be a very beneficial addition to the paper, as expressed above, we feel that the full exploration and quantification of the observational uncertainty present in the MetUM model (as well at the observations themselves) is beyond the scope of this study, and we are unsure of how to translate the RMS difference into confidence intervals. We do, however, agree that the effect of the model bias on the results of the analysis should be explored. We have therefore reproduced the plots in Fig. 2 (a) and Fig. 3 (a) of the original manuscript using GSOD data for stations within the London and Amsterdam model grid cells. We have added this plot in the supplementary material and refer to it at the end of Section 3.2:*

As noted in Section 2, the discrepancy in model generated wind-gust speeds compared to observations could lead to differences in the identification of the extremal dependence class between locations. To explore this discrepancy we extract daily maximum wind gust speeds at Amsterdam Schiphol Airport (weather station within the Amsterdam grid cell) from the Global Summary of the Day (GSOD) data set, and calculate windstorm footprint maximum wind gust speeds as was done for London City. [Figure 2 in attachments] (a) in the Supplementary Material presents a comparison of these footprint maximum wind gust speeds for London City and Amsterdam Schiphol Airport, the observation equivalent of Fig. 2 (a), and [Figure 2 in attachments] (b) in the Supplementary Material presents the empirical extremal dependence coefficient $\chi(p)$ for $p \in [0, 0.4]$ calculated for this pair of observation stations, the observation equivalent of Fig. 3 (a). This comparison indicates that footprint wind gust speeds at the London and Amsterdam weather stations are, in fact, less related in the extremes since, the most extreme events in each location do not coincide. This is reflected in the empirical extremal dependence coefficient in [Figure 2 in

attachments] (b), which deceases more rapidly than in Fig. 3 (a) as $p \to 0$. This difference may be due to the underestimation of extreme wind gust speeds in the MetUM model, as discussed in Section 2 and Roberts et al. (2014), resulting in the modelled footprints missing the most extreme winds and hence producing wind gust speed that are more similar, and less extreme, in both locations. The empirical indication of asymptotic independence is, however, consistent for both observations and MetUM modelled windstorm footprints.

[Figure 2 in attachments] (a) Scatter plot comparing observed windstorm event maximum wind gust speeds (ms$^{-1}$) at London City and Amsterdam Schiphol Airport GSOD weather stations, (b) extremal dependence measure $\chi(p)$, for $p \in [0, 0.4]$, for observed windstorm footprint wind gust speeds at London City GSOD weather station paired with Amsterdam Schiphol Airport GSOD weather station.

**0.2 Events analysed in Figure 4, and interpretation of results**

- **Fig 2 indicates approx. 25 points above quantile=0.99, which suggests that the quantile=0.5 in Figure 4 is based on over one thousand storm events in a 35 year period. The inclusion of about 30 events per year on average will contain many breezy days. These data points are potentially misleading to include, because the spatial structure of days with weak winds is likely to be substantially different from the spatial structure of severe events producing tail winds. I request that Figure 4 is re-drawn using data from quantile=0.9 and upwards. This would still include weak winds from minor cyclones, but is a step in the right direction, while maintaining sufficient sample sizes. The conclusions to be drawn from Figure 4a should be reviewed in a revised version of manuscript. First, the results in Figure 4a indicate rising values of the coefficient of tail dependence for quantile**

**thresholds above 0.9, towards a value of unity for the highest quantile. Given the aim to capture behaviour in the limit as p tends to 0, it seems unsafe to conclude that London-Amsterdam has tail independence. Second, the inclusion of observational uncertainty (point 1 above) will broaden the confidence limits which may require a new interpretation of results.**

*Thank you for this feedback, however we feel there must be some confusion in the interpretation of the coefficient of tail dependence ($\eta$). This parameter is equivalent to the scale parameter of a Poisson process (or the shape parameter of a GPD (see Ledford and Tawn (1996)). Therefore $\eta$ is a parameter of a model for the joint excesses of the pair which determines the asymptotic behaviour, and should be chosen such that the model is stable above the threshold (similar to when choosing the threshold in the Generalised Pareto Distribution (GPD) model), not an asymptotic measure of dependence like $\chi$ and $\bar{\chi}$, which we are interested in as $p \to 0$.*

*We have now made this interpretation of $\eta$ clearer by editing the paragraph after Eqn. (4):*

We fit this model to the pairs London-Amsterdam and London-Madrid, requiring the specification of the high threshold, $w$, above which the Poission process model is fit. As discussed by Ferro (2007), this threshold selection is a trade-off between being low enough that enough data is attained to ensure model precision, but high enough that the extreme-value theory that motivates the model provides accurate estimates, suggesting we should select the lowest level at which the extreme-value approximations are acceptable (Ferro (2007)). In a similar way to choosing the appropriate threshold when fitting a Generalised Pareto Distribution (see Coles (2001)), empirical diagnostic plots can be used to inform this selection. For example parameter stability plots, in which the

estimated model parameters and mean excess should be constant above the chosen high threshold; and quality of fit plots, in which for this model, the transformed excesses, $(Z - w)/\eta$, should be exponentially distributed if an appropriately high threshold has been chosen (see Ferro (2007) for more details).

Here, the $85\%$ quantile of the structural variable $T$ is selected, based on these diagnostic plots (examples for these plots for London-Amsterdam are presented in [Figure 3 in attachments] in the Supplementary Material). This choice is similar to the $0.88\%$ and $0.9\%$ thresholds selected in the applications of Ferro (2007) and Ledford and Tawn (1996) respectively.

[Figure 3 in attachments] Threshold selection diagnostic plots for the Ledford and Tawn (1996) model: (a) Quantile-Quantile plot comparing the transformed excesses, $(Z - w)/\eta$, with the Exponential distribution (with rate parameter equal to 1), where $w$ is the selected 0.85% quantile of the structure variable $T$, (b) stability plot for the mean excess as a function of threshold $w$.

*The diagnostic plots in Fig. 4 are equivalent to those shown in Figures 3 and 4 in Ledford and Tawn (1996) and Figure 2 in Ledford and Tawn (1997). In both of these papers they use the range (0.5-1). We agree that a threshold as low as 0.5 is most likely too low to give an appropriate extreme-value theory model, however, we would like to keep this range on the plots to be in line with the aforementioned papers. We have added a clarification of this replication of their approach by editing the paragraph after Fig. 4:*

As in Ledford and Tawn (1996, 1997), here this is done by observing the proportion of time $\eta = 1$ is within the profile likelihood confidence interval for $\eta$, when estimated using $w$ in the interval of $0.5 - 1$ quantile of $T$. The pair

$(X, Y)$ are said to be asymptotically dependent if $\eta = 1$ is contained within these confidence intervals for a majority of the range of $w$, and asymptotically independent otherwise.

**0.3 Section 3.4 conjecture**

- **The conjecture to explain the tail independence in section 3.4 begins by representing storm winds as isotropic turbulence. It is standard to represent storm winds as the sum of a mean wind and a smaller turbulent contribution. This is also consistent with the MetUM model gust dataset used by the authors (description around Line 124). The authors then assume that gusts at two locations are bi-variate normal. While the turbulent contribution to winds at two distinct locations might be bi-variate normal at any instant in time, the gusts analysed for tail dependence are the maximum gust over the whole storm. The storm-max gusts between neighbouring locations are expected to have strong tail dependence since they would have very similar mean wind and max gustiness from isotropic turbulence.**

*Thank you for raising this interesting issue, which we had not addressed in the turbulence discussion. This paragraph of explanation has now been added after the first paragraph in Section 3.4:*

It is useful to first consider the more tractable problem of dependency in simultaneous wind speeds rather than maximum wind speeds over a given time period. The dependency between maximum gust speeds over 3 days will not generally be less than the dependency between simultaneous wind gust speeds because maximum wind gusts for a storm do not occur at the same time at different locations. However, for locations that are close to one another,

maximum gust speeds for fast moving extreme storms will occur within a short time window (e.g. within around 3 hours or less for extreme storms over the UK) and so simultaneous results become more relevant.

- **Regarding the text on Lines 254-258: McNeil et al. (2005) showed that if correlation is less than one, then the coefficient of upper tail dependence equals zero, their Example 5.32. McNeil, A J, Frey, R, and Embrechts, P: Quantitative Risk Management Concepts, Techniques, Tools. Princeton University Press, 2005**

  *Thank you for this reference. We have now added this in line 274:*

  So what can be deduced about the extremal dependence class of wind speeds from such turbulence models? Firstly, as shown in Example 5.32 of McNeil et al. (2005), since the individual velocity components are bivariate normal, the individual velocity components are asymptotically independent at different locations e.g. $u_1 = u(s_1)$ and $u_2 = u(s_2)$ are asymptotically independent when $s_1$ differs from $s_2$, and likewise for $v(s)$.

0.4   Section 4 on losses

- **Lines 277-283: the authors state a simple loss function provides better storm loss estimates than the Klawa and Ulbrich (KU) loss function. There are various reasons why this judgement on loss functions is misleading. Besides the minor fact that the two articles quoted excluded population weighting hence did not test the KU loss function, there is a more significant issue that 'better' is defined in non-standard and highly specific**

terms as 'a subset of 23 significant storms completely contained in highest quantile of all storms'. Further, there is much published work on how loss severity is a function of wind speed, and KU's loss function certainly captures this effect more accurately than a step function.

The 'conceptual loss function' used by the authors could be more accurately described as areal frequency of loss occurrence, and ignoring loss severity. Its usefulness in estimating total loss is an interesting result, since it suggests the area of storm above a loss-causing threshold is the dominant contributor to total storm loss. I suggest the authors describe their loss function in more specific terms as 'areal frequency of loss' in the text.

Further, if the authors wish to retain text comparing a step function to the superior KU loss function, then the authors should include more information for readers: errors in loss estimates depend on wind speeds, loss functions and exposure density, and the success of the simplest loss function over KU in tests performed by Roberts et al. is very likely due to its relative insensitivity to errors in other components of their loss estimates, such as estimated gusts. This helps resolve the dilemma of a rapid growth of loss with windspeed indicating KU, while a less sophisticated testing framework indicated a step function.

*We agree that this introduction of the loss function needs to be more detailed and balanced. In combination with this comment and those of Reviewer 1, we have decided to restructure Section 4 and introduce a more generic form for the loss function. In doing so, we discuss the KU loss function along with other functions and give a more detailed justification of our chosen loss function, incorporating your comments and concerns as caveats. After the first paragraph*

*of Section 4 we have edited:*

Similar to other natural hazard loss models, in the absence of confidential insurance industry exposure and vulnerability information, it has become common in the literature to define conceptual windstorm loss as a function of the footprint wind gust speeds (see Dawkins et al. (2016) for a review). While these conceptual windstorm loss functions vary in the detail of their composition, it is possible to express most in a general form, for the pair $(X, Y)$, as:   L(X,Y) = g[V(X)e(X)H{X-U(X)} + V(Y)e(Y)H{Y-U(Y)}]  where $V$ is a function the wind gust speeds characterising the magnitude of the hazard, $e$ represents exposure (e.g. population density), $U$ quantifies a high threshold of the wind gust speed above which losses occur, $H$ is a Heaviside function such that $H\{m\} = 1$ if $m > 0$ and $H\{m\} = 0$ otherwise, and $g$ is an additional function applied in some cases to reduce skewness.  For example, in the widely used and rigorously validated conceptual loss function of Klawa and Ulbrich (2003), $V(X) = (X - x_{0.98})^3$, $U(X) = x_{0.98}$ (where $x_{0.98}$ is the $98\%$ quantile of $X$) and $e(X)$ is represented by the population density at the location (with equivalent expression for $Y$), while Cusack (2013) used a loss function in which $V(X) = (X - x_{0.99})^3$, $U(X) = x_{0.99}$, the $99\%$ quantile of $X$, and $g[\cdot] = \sqrt[3]{\cdot}$.  See Table 2.1 in Dawkins (2016) for a summary of previously published conceptual loss functions in terms of the components of Eqn. 0.4.

More recently, Roberts et al. (2014) presented an exploration of the success of a number of these conceptual windstorm loss functions in representing insured loss throughout the European domain, based on the same data set as in this study, with the aim of developing a method for selecting extreme storms for the eXtreme WindStorms (XWS) catalogue. While there is much published work on the existence of a relationship between loss severity and the magnitude of the wind, in particular the cubed excess wind as used in the loss functions of

Klawa and Ulbrich (2003) and Cusack (2013), Roberts et al. (2014) found that a conceptual loss function representing just the area in which the windstorm footprint exceeds a high loss threshold (i.e. $V(X) = 1$ and $e(X) = 1$ in Eqn. 0.4) to be more successful at characterising a subset of extreme windstorms known to have caused large insured losses. It should be noted however, that this exploration did not include population density within the Klawa and Ulbrich (2003) loss function, and was therefore not a direct comparison of this measure. In addition, an alternative subjectively selected subset of extreme storms may have given an alternative result, and the success of this simplistic 'areal frequency of loss' function in representing losses in this climate model generated data set of windstorm footprints may be due to its relative insensitivity to errors in other components of the loss estimates, such as estimated gusts, and may not perform as well as other loss functions if applied to wind gust observations.

However, following the results of Roberts et al. (2014) in the context of this data set, and in line with Dawkins et al. (2016), within this study we propose a similar threshold exceedance conceptual loss function. Roberts et al. (2014) used an exceedance threshold of 25ms$^{-1}$ while Dawkins et al. (2016) used a threshold of 20ms$^{-1}$, as is commonly used by German insurance companies (Klawa and Ulbrich (2003)). Here, similar to Klawa and Ulbrich (2003) and Cusack (2013), we propose a locally varying wind gust speed quantile threshold, accounting for local adaptation to varying wind intensity.

- **The over-estimation of joint loss probabilities in the maps in Figures 7e & f are explained as a mis-specification of asymptotic dependence (lines 308-309). However, it could be due to a too high estimate of the dependence parameter r. Could the authors include in the text the group of**

**data used for estimating the dependence parameter?** *We have now edited the paragraph beginning on line 223 in the original manuscript to include this information:*

The Gumbel bivariate copula model characterises asymptotic dependence with the degree of dependence quantified by parameter $r$. For each pair of locations, this parameter is estimated via maximum likelihood using the `copula` R package. The Gaussian bivariate model characterises asymptotic independence with dependence parameter $\rho$, here, for each pair of locations, represented by the Spearman's rank correlation coefficient. Both models are fit to the full bivariate data pair, as presented in Fig. 2. For the Gumbel model the data is transformed to uniform margins using the empirical distribution function. The same transformation is made for the Gaussian model, followed by a transformation to Gaussian margins using the standard normal distribution function. The parametric forms of $\chi(p)$ and $\bar{\chi}(p)$ for these two opposing models are shown in Table 1. In Fig. 3, the Gumbel model is calculated as in Table 1, however, since the closed form definition for the Gaussian model in Table 1 only holds for the limit $p \to 0$, for this model $\chi(p)$ and $\bar{\chi}(p)$ are estimated as the median of 1000 parametric bootstrap simulations from the associated bivariate normal distribution.

0.5   Technical Corrections

- **There are many instances of 'apposing' when 'opposing' may be more appropriate?**

  *Thank you, we have now corrected these.*

**References**

Coles, S. (2001). *An Introduction to Statistical Modeling of Extreme Values*. Springer.

Cusack, S. (2013). A 101 year record of windstorms in the Netherlands. *Climate Change*, 116:693–704.

Dawkins, L. C. (2016). *Statistical modelling of European windstorm footprints to explore hazard characteristics and insured loss*. PhD thesis, University of Exeter, College of Engineering Mathematics and Physical Sciences.

Dawkins, L. C., Stephenson, D. B., Lockwood, J. F., and Maisey, P. E. (2016). The 21st century decline in damaging european windstorms. *Natural Hazards and Earth System Science*, 16:1999–2007.

Ferro, C. A. T. (2007). A probability model for veryfiying deterministic forecasts of extreme events. *Weather Forecasting*, 22:1089–1100.

Klawa, M. and Ulbrich, U. (2003). A model for the estimation of storm losses and the identification of severe winter storms in Germany. *Natural Hazards and Earth System Sciences*, 3:725–732.

Ledford, A. W. and Tawn, J. A. (1996). Statistics for near independence in multivariate extreme values. *Biometrika*, 83(1):169–187.

Ledford, A. W. and Tawn, J. A. (1997). Modelling dependence within joint tail regions. *Journal of the Royal Statistical Society*, 59(2):475–499.

McNeil, A. J., Frey, R., and Embrechts, P. (2005). *Quantitative Risk Management: Concepts, Techniques, Tools*. Princeton University Press.

Roberts, J. F., Dawkins, L., Youngman, B., Champion, A., Shaffrey, L., Thornton, H., Stevenson, D. B., Hodges, K. I., and Stringer, M. (2014). The XWS open access catalogue of extreme windstorms in Europe from 1979 to 2012. *Natural Hazards and Earth System Science*, 14:2487–2501.

Sheridan, P. (2011). Review of techniques and research for gust forecasting and parameterisation: Forecasting research technical report 570. Technical report, Met Office.
* * *
[Figure]

**Fig. 1.**

[Figure]

**Fig. 2.**

[Figure]

**Fig. 3.**

---

## Author Comment (AC3) · 10 Aug 2018

Figures associated with Response to Reviewer 1

[Figure]

**Fig. 1.**

[Figure]

**Fig. 2.**

[Figure]

[Figure]

**Fig. 3.**

[Figure]

NHESSD

Interactive
comment

**Fig. 4.**

[Figure]

---

## Author Comment (AC4) · 10 Aug 2018

Key:

- **Reviewer's comment**

- *Our response*

- Additional/edited text in the manuscript

[Figure]

**0.1 Main Comments**

- **1. a. The authors repeatedly claim the novelty of their approach (ll. 63, 100, 103). As they implicitly note on ll. 63, the main novelty lies in the combination of existing modelling approaches rather than in some fundamental statistical advance. However, conceptually very similar approaches for investigating the appropriate dependence class for spatially remote geophysical extreme events have been implemented before, within a more comprehensive theoretical framework (for example, see Kereszturi et al. 2016). Other than being applied to a different variable, what broad additional insights does the present study provide?**

  *Thank you for this comment and very relevant citation. Since reading and responding to all reviewer comments, and reading the suggested literature, we have decided to demonstrate the motivation and novelty of this paper in a new way. In line with the natural hazards theme of the journal, we will focus on developing an approach for, firstly, systematically exploring the dominant extremal dependence class throughout a high dimensional continent wide data set (e.g. windstorm footprint), relevant for the catastrophe modelling of a diverse insurance portfolio for a continent wide natural hazard, and secondly, relevant for natural hazards, how this extremal dependence specification effects the representation of insurance losses. In addition, this loss representation is proposed as an additional, natural hazards relevant, diagnostic for the extremal dependence class. We have not seem any examples in the literature of using the extremal dependence measure of Ledford and Tawn (1996) and Coles (2001) to systematically explore very high dimensional data, nor of bringing this comparison through to natural hazard aggregate losses. Throughout the paper we now use the London-Amsterdam, London-Madrid pairs to introduce the extremal dependence measures and then present a systematic approach for*

*using the same measures to explore the extremal dependence throughout the high dimensional domain (see response to comment 2 below for this additional analysis). We have rewritten the introduction to reflect this change (see end of comment 1 for revised Introduction).*

- **b. On a related note, the authors suggest that an important result of their work will be to simplify the development and use of models that correctly represent extremal dependence for the variable of interest, removing the need to apply more complex - but more flexible - models which account for the different possible dependence classes (ll. 91-95). There are a number of these models available, including those of Wadsworth et al. (2017) and Huser and Wadsworth (2018). The actual benefits of the approach proposed by the authors are not explicitly described in the manuscript. Are the authors suggesting that the final result stemming from their approach outperforms these models (or that the results are comparable but require less work?) If so a comparison should be provided. Or that the reduction in computational time is so large as to make a difference in practical applications (if so, some indicative figures should be provided)? Or that the ease of implementation of their approach makes it applicable to datasets where other models couldn't be applied? Again, some examples should be provided and the extent/range of validity of this advantage should be discussed. Any one of the above points would be a sound motivation for the present work, but they would need to be explicitly stated and factually supported.**

*Thank you, this is an important point and we agree the motivation for the work needs to be clearer and based within the context of the relevant literature.*

*As described above, in response to the comments received this motivation has been proposed in a different way.*

*It is our understanding that, while there is a growing literature in the area of flexible models for extremal dependence which can accommodate higher and higher dimensional data, all such models are still limited by dimensionality. For example Huser and Wadsworth (2018) identify that their model is only feasible in moderate dimensions and note that, with the exception of the specific model used in de Fondeville and Davison (2018), truly high-dimensional inference for spatial extreme-value models has yet to be achieved. Indeed, as noted by de Fondeville and Davison (2018), this dimensionality limitation is true for max-stable models.*

*Following on from the described re-contextualisation in the above comment, here we aim to model very high dimensional ( 15000 locations) windstorm hazards data, relevant when modelling natural hazards that effect a large spatial domain (e.g. a whole continent). Therefore, we argue that the application of these flexible models is computational infeasible and instead we must use a systematic diagnostic approach to identify the dominant extremal dependence class throughout the high dimensional data domain, and model the full domain based on this dominant characteristic (e.g. using the model of de Fondeville and Davison (2018) if asymptotically dependent or a geostatsticial Gaussian process or the Gaussian tail model of Bortot et al. (2000), if asymptotically independent). If the high dimensional data characterises both asymptotic dependence and asymptotic independence at different separation distance, we suggest that the conceptual aggregate loss diagnostic can be used to explore how and where this mis-specification of extremal dependence effects the modelled output of interest (the loss).*

*To achieve this we use the bivariate measures in the paper as they are quick to estimate and can therefore be used to explore many thousands of pairs of locations, giving a detailed understanding of the high dimensional data.*

*We have rewritten the introduction to reflect these comments (see end of responses to comment 1).*

- **c. As a final note, very little is said in the introduction of the above-mentioned models which account for a broad range of dependency classes (see also references in Huser et al., 2017). There is a growing literature in this subfield, which should be discussed. With the above I don't suggest that the work of the authors is devoid of interest, but they should certainly explain more clearly what the real novelty of the study and what the advantages it will provide to the community are. In my view, it will not be sufficient to alter one or two sentences in the manuscript: this will require a substantial clarification and contextualization effort, and likely some additional analysis to support the claims made.**

*We agree with this comment and have now included a thorough review of these very relevant papers within our introduction. As described above, we have re-contextualised the paper to have two clear novelties, relevant for the natural hazards community, 1 - an approach for systematically exploring extremal de-pendence in very high dimensional natural hazards data (relevant for modelling wide spread impact), necessary since flexible models for extremal dependence are limited by dimensionality, 2 - understanding how specification of extremal dependence class effects the hazard model representation of insured loss using an additional, natural hazards relevant, conceptual loss extremal dependence diagnostic approach. In implementing this re-contextualisation we have rewritten the introduction to describe these aims and reviewed the relevant literature (see*

*below), developed and applied approaches for systematically exploring the high dimensional domain, requiring substantial additional analysis (see response to comment 2), and introduced a more generic conceptual loss function for broader applicability (see responses to Reviewers 1 & 2). We feel that attempting to apply the approach of Huser and Wadsworth (2018), for example, would be irrelevant here since the aim is to eventually model the full high dimensional data, and rather a thorough discussion of the merits and limitations of such approaches is adequate when combined with the large alterations made to the motivations, methodologies and scope of the paper.*

*The rewritten Introduction:*

[revised manuscript text omitted]

The remaining paper is organised as follows. The windstorm hazard dataset used throughout this paper, is described in Section 2. In Section 3 we introduce and apply the extremal dependence diagnostics of Ledford and Tawn (1996) and Coles et al. (1999), firstly to two pairs of locations and secondly to systematically explore the high-dimensional data domain, and contribute a physical explanation for the form of extremal dependence identified in the windstorm hazard fields. Section 4 describes our additional, natural hazards relevant, conceptual aggregate loss extremal dependence diagnostic approach, and finally, Section 5 concludes.

- **2. My second major concern regarding this study is the fact that the results are presented only for two location pairs (with one location common to both). The authors briefly mention the fact that they have tested their results for other locations (ll. 250-252), but this is not substantiated in any meaningful way. Is there a way to systematically test the robustness of the results obtained by the authors across a western European domain, perhaps presenting the results in a form similar to Fig. 7 but for different reference locations or a 2-D version of Fig. 8 showing location on one axis,**

**conditional joint loss on the other and density as colours/contours?**

*Thank you, this is a very important point, especially since now one contribution of the paper is an approach for exploring extremal dependence in very high dimensional data. This point was also made by Reviewer 1, in the context of identifying the dominant class of extremal dependence throughout the domain. We have added this part of the additional analysis at the end of Section 3.3:*

When aiming to develop a statistical model for high dimensional spatial data over a large geographical domain, it is essential to systematically explore the dominant extremal dependence class across all locations. Here, we present an approach for doing so, which uses this quick-to-calculate coefficient of tail dependence diagnostic, demonstrated by application to our windstorm footprint data set. We first take a stratified (based on the distribution of locations over longitude and latitude) sample of 100 locations within the European domain. One such sample is shown in [Figure 1 in attachments](a). Since the extremal dependence is likely to decrease with increasing separation distance (Wadsworth and Tawn (2012)) and we hope to understand if asymptotic independence is dominant and hence present at small separation distances, for each of these 100 locations, we estimate the coefficient of tail dependence, $\eta$ (and the associated 95% profile likelihood confidence interval) when paired with the 100 nearest locations within the full domain. [Figure 1 in attachments](b) demonstrates how the 100 nearest locations are geographically distributed for one such sampled location in our windstorm footprint dataset. For each pairing, the coefficient of tail dependence is calculated using w as the 0.9 quantile threshold of the structure variable, found to ensure stable estimates of $\eta$ using diagnostic plots as in [Figure 1 in attachments] (c). The estimated $\eta$ parameters and confidence intervals for these $100{\times}100$ pairs of locations are plotted against separation distance to explore how, throughout the domain, $\eta$ varies at small separation

distances and changes with increasing separation distance, shown in [Figure 1 in attachments] (d). The parameter estimate related to the pair of locations in pink and blue in [Figure 1 in attachments] (b) is shown in pink. This method is repeated many times with 10 such repetitions shown in [Figure 1 in attachments] of the Supplementary Material at the end of the paper, showing very similar results.

[Figure 1 in attachments here] - (a) A stratified (based on the distribution of locations over longitude and latitude) sample of locations within the European domain, with stratified grid shown in grey; (b) a demonstration of the 100 nearest locations [turquoise] to one of these sampled locations [blue], with one such point selected at random [pink]; (c) the coefficient of tail dependence diagnostic plot (as in Fig. 4) for wind gusts at the blue location paired with the pink location; (d) the coefficient of tail dependence (estimated using w as the 0.9 quantile threshold of the structure variable) and 95% profile likelihood confidence intervals, for each of the 100 sampled locations paired with their 100 nearest locations in the full domain, plotted against separation distance in kilometres, with the estimate based on the pair of locations in (b) and (c) added in pink.

[Figure 1 in attachments] (d) shows that for small separation distances ($<180$ km) a proportion of pairs of locations have coefficients of tail dependence parameter, $\eta$, estimates close to 1, with $\eta = 1$ within the confidence interval, indicating asymptotic dependence. Within the range (0-50 km) 69% of pairs of locations exhibit this behaviour, however this proportion reduces rapidly as separation distance increases, to 30% for locations separated by (50-100 km), 13% for locations separated by (100-150 km) and 3% for locations separated

by (150-200 km). Hence, while there is evidence of asymptotic dependence for some locations in close proximity, even at very small separation distances (50 km) a larger proportion of locations exhibit asymptotic independence. Indeed, here and in [Figure 2 in attachments] of the Supplementary Material, beyond a separation distance of approximately 200km the coefficients of tail dependence parameter estimates drop well below 1, indicating asymptotic independence. Therefore, since separation distances within the domain extend to up to 3500km, we conclude that asymptotic independence is the dominant extremal dependence structure across the spatial domain.

It is important to consider the validity of representing even this small proportion of asymptotically dependent pairs of locations incorrectly as asymptotically independent. To explore this, Bortot et al. (2000) carried out a simulation study in which they fit the Gaussian, Ledford and Tawn (1996) and Gumbel models to bivariate data simulated from three parent populations with different classes of extremal dependence. They conclude that, for asymptotically independent parent populations the Gaussian copula is able to provide accurate inferences for tail probability estimates, out performing the Gumbel copula model, and even for asymptotically dependent parent populations, the estimation error of the Gaussian copula model was deemed to be acceptably small. This suggests that, when data dimensionality prohibits the use of flexible extremal dependence models, such as Huser and Wadsworth (2018), and asymptotic independence is found to be the dominant extremal dependence structure across the spatial domain, using an asymptotically independent model, such as the Gaussian tail model, is preferable over using a model for asymptotic dependence throughout the domain. In Section 4 we present a further, natural hazards relevant, diagnostic approach for further validating this, based on an estimates of the aggregate natural hazard losses.

[Figure 2 in attachments here] - For 10 stratified samples of 100 locations within the European domain: the coefficient of tail dependence (estimated using w as the 0.9 quantile threshold of the structure variable) and 95% profile likelihood confidence intervals, for each of the 100 sampled locations paired with their 100 nearest locations in the full domain, plotted against separation distance in kilometres.

*We have also added an equivalent systematic, domain wide, comparison for the conceptual loss part of the paper. Firstly, as an extension of Fig. 7 in the manuscript, by plotting the bias in modelled $\chi(0.1)$, when the Gaussian and Gumbel bivariate models are fit to a stratified sample of 100 location paired with the other 99 locations, against separation distance. Secondly, as an extension of Fig. 8, as you have suggested, a 2-D version in which location is shown on the x axis, expected conditional loss on the y axis, and the difference between the empirical and modelled densities is coloured. This plot is created based on a further stratified sample of 100 location paired with all others in the domain. We have included this first additional analysis and plot after Fig. 7:*

We extend this analysis to systematically explore the high-dimensional domain by fitting both the Gaussian and Gumbel models to a stratified sample of 100 locations paired with each of the other 99 locations, and, for each pair, plot the difference between empirical and modelled $\chi(0.01)$ against their separation distance, shown in [Figure 3 in attachments]. This domain-wide comparison indicates that, while the Gaussian model slightly over and under estimates empirical $\chi(0.01)$ at small separation distances, this model consistently outperforms the Gumbel model, which overestimates $\chi(0.01)$ for all separation distance, even very small. This indicates that a majority of nearby locations do not exhibit asymptotic dependence as they are not well represented by the Gumbel

model, further supporting the diagnosed dominance of extremal independence throughout the domain of our dataset.

[Figure 3 in attachments here] - The difference between empirical and modelled $\chi(0.01)$ for a stratified sample of 100 locations paired with each of the other 99 locations, plotted against separation distance for (a) the Gaussian model and (b) the Gumbel model.

*We have then added the second additional plot and analysis after Fig. 8:*

Again, we extend this analysis to systematically explore the robustness of these results throughout the high-dimensional domain. To achieve this we carry out the same calculation as in Fig. 8, replacing London as the location of origin, with each location within a stratified sample of 100 locations. For each of these 100 locations, [Figure 4 in attachments] presents the the difference between modelled and empirical relative frequencies of binned ranges of conditional expected joint loss, separately for the Gaussian and Gumbel models, i.e. representing the difference between the modelled and empirical density plots in Fig. 8, but for 100 locations rather than one. [Figure 4 in attachments] (b) identifies that the discrepancy between the empirical and Gumbel estimates of conditional expected joint loss shown in Fig. 8 are consistent throughout the domain, with lower values being under-represented and higher values, even as high as 0.014, over-represented by the Gumbel model. In a similar way, [Figure 4 in attachments] (a) shows that the Gaussian model performs equally well for these 100 locations, with much smaller discrepancy compared to the Gumbel model, as found in Fig. 8.
[Figure]

[Figure 4 in attachments here] - For the 100 sampled locations shown in [Figure 1 in supplementary material] (a), the difference between modelled and empirical relative frequencies of binned ranges of expected conditional joint loss shown on the y axis, for (a) Gaussian model, (b) Gumbel model.

This novel conceptual aggregate loss diagnostic approach supports the use of the Gaussian model when asymptotic independence is found to be the dominant extremal dependence characteristic within a high dimensional natural hazards dataset. In this windstorm footprint application, while the Gumbel model is able to represent some pairs of locations at very small separation distances, where asymptotic dependence is suggested by the coefficient of tail dependence, this model greatly misrepresents the joint tail behaviour and hence the conditional probability of joint loss for a majority of pairs and separation distances. Conversely, the Gaussian model is able to represent the joint tail behaviour relevant for loss estimation for locations within close proximity to each other, as well as further apart.

**0.2 Additional Comments**

- **3. The title suggests a very broad relevance of the paper. Even though the techniques discussed in the study are general, the analysis effectively focusses on windstorms at three specific locations. As such, the current title is misleading and should be changed to reflect the contents of the study. Alternatively, the approach proposed by the authors should be applied to other geophysical variables and geographical domains.**

  *We have now altered the title of the paper to more closely reflect the contributions of the re-contextualised paper and the specific windstorm application,*

*to:*

Quantification of extremal dependence in spatial natural hazard footprints: Independence of windstorm gust speeds and its impact on aggregate losses

- **4. l. 124: Please include a reference for how the wind gusts are calculated. This parametrisation is very simple. If it works well, simplicity is obviously good, but a brief discussion of its performance versus alternative approaches should be provided.**

  *We have now added two references to the introduction of the paramterisation in Section 2:*

  As described by Roberts et al. (2014), the wind gust speeds are calculated from wind speeds in the MetUM model, based on a simple gust parameterisation $U_{gust} = U_{10m} + C\sigma$, where $U_{10m}$ is the wind speed at 10 metre altitude and $\sigma$ is the standard deviation of the horizontal wind, estimated from the friction velocity using the similarity relation of Panofsky et al. (1977). The roughness constant $C$ is determined from universal turbulence spectra and is larger over rough terrain.

  *We then, in combination with a response to Reviewer 2, include a few paragraphs at the end of Section 2 to discuss this parameterisation and the general validity of the MetUM modelled footprints:*

  Using model generated windstorm footprints for representing historical storms has benefit in terms of spatial and temporal coverage, however these estimated

maximum wind-gust speeds will inevitably differ from the those observed at nearby weather stations. For example, as noted by Roberts et al. (2014), several alternative methods for parameterising wind gust speeds are available (see Sheridan (2011) for a review), which can lead to large differences in estimated gusts (10-20ms$^{-1}$). The validity of simplistic gust parameterisation stated above was evaluated by Roberts et al. (2014), who found an overestimation in the effect of surface roughness at stations greater than $\sim$ 500 metre altitude, leading to underestimation of MetUM modelled extreme winds in these locations. In addition, within this thorough evaluation of MetUM windstorm footprints, Roberts et al. (2014) identified a slight underestimation in extreme wind gust speeds greater than $\sim 25$ms$^{-1}$. This was found to be due to a number of mechanisms including the underestimation of convective effects and strong pressure gradients, leading to the underdevelopment of fast moving storms (Roberts et al. (2014)).

- **5. Section 3.4: Are wind gust speeds really independent Gaussian processes? Can this be tested on the data available to the authors?**

*The reviewer raises a good point. A brief justification has been added to Section 3.4:*

[revised manuscript text omitted]

**(a)**

**(b)**

**(c)**

**(d)**

**Fig. 1.**

[Figure]

**Fig. 2.**

[Figure]

**Fig. 3.**

[Figure]

[Figure]

**Fig. 4.**

[Figure]

**Fig. 5.**